# Granger-Causal Hierarchical Skill Discovery

## Abstract

Reinforcement Learning (RL) has shown promising results learning policies for complex tasks, but can often suffer from low sample efficiency and limited transfer. We introduce the Hierarchy of Interaction Skills (HIntS) algorithm, which uses learned interaction detectors to discover and train a chain hierarchy of skills that manipulate factors in factored environments. Inspired by Granger causality, these unsupervised detectors capture key events between factors to sample efficiently learn useful skills and transfer those skills to other related tasks—tasks where many reinforcement learning techniques struggle. We evaluate HIntS on a robotic pushing task with obstacles—a challenging domain where other RL and HRL methods fall short. The learned skills not only demonstrate transfer using variants of Breakout, a common RL benchmark, but also show 2-3x improvement in both sample efficiency and final performance compared to comparable RL baselines. Together, HIntS demonstrates a proof of concept for using Granger-causal relationships for skill discovery in domains where a chain of skills can represent complex control.

## 1 Introduction

Reinforcement learning (RL) methods have shown promise in learning complex policies from experiences on various tasks, from weather balloon navigation Bellemare et al. (2020) to Starcraft Vinyals et al. (2019). Nonetheless, they often struggle with high data requirements and brittle generalization in their learned controllers Nguyen & La (2019).

To address the limitations of vanilla RL, hierarchical RL (HRL) methods exploit temporal and state abstractions. Unlike standard "flat" RL methods that aim at learning a monolithic policy, HRL constructs a temporal hierarchy in which higher-level policies invoke subsequent lower-level policies (also called *skills* or *options*) and the lowest-level skills invoke primitive actions. This hierarchical structure offers three major benefits: First, skills can represent temporally extended behaviors, lasting multiple steps and decomposing long-horizon tasks into a sequence of shorter temporal segments. Second, an appropriate selection of skills fosters useful state abstractions, reducing the state-space complexity by merging states based on their behavioral characteristics (*e.g.*, their preconditions and effects). Third, the lower-level skills can be reused across different tasks to enable multi-task learning and fast adaptation Kroemer et al. (2019).

Skills can be acquired through either reward-based methods, which propagate reward information to acquire the skill behavior, or reward-free methods, which often use state-covering statistics Eysenbach et al. (2018) to learn distinct skills. While a handful of methods suggest ways to distinguish visiting one state over another Şimşek & Barto (2008), many reward-free methods struggle in high-dimensional environments because learning skills to cover the entire state space is inefficient. Furthermore, even after learning those skills, the number of skills can end up being too large for a high-level controller to explore efficiently. On the other hand, reward-based skill learning Konidaris & Barto (2009); Levy et al. (2017b) propagates information from the rewards only when the agent achieves meaningful progress. For long-horizon tasks, it often means that the learner flounders indefinitely before the first success is observed, stalling learning.

While HRL methods have shown promising results, the floundering and task-specific limitations in reward-based methods and inability to scale to large state spaces in reward-free unsupervised methods result in low sample efficiency and transfer, preventing both from realizing the benefits of HRL. This work introduces a

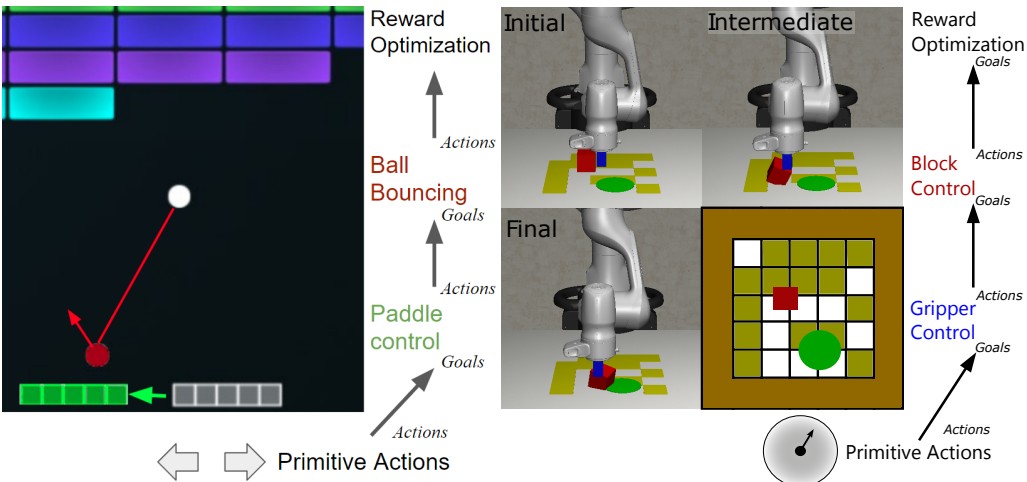

Figure 1: **Left:** The chain of HIntS goal-based skills for Breakout, from primitive actions to final reward optimization. The goals of one factor is the action space of the next factor in the chain. HIntS uses Granger-causal tests to detect interactions and construct edges between pairs of factors. **Right:** The Robot Pushing domain with negative reward regions. The objective is to push the block (red) from a random start position to a random goal (green) while avoiding the negative regions (shown in yellow, −2 reward), which are generated randomly in 15 grid spaces of a 5×5 grid over the workspace.

novel reward-free method for skill discovery in factored state spaces drawn from human cognition: learning skills that cause factors to interact. In this work, we use "factor" to describe both a collection of features like the state of an object, to non-state factors such as actions and rewards. Humans, even as young as infants, exhibit exploratory behavior without top-down rewards. Such behavior is not state-covering but directed towards causing particular effects between factors Rochat (1989). Exploiting this causal framework decouples a small subset of interacting factors instead of the state space involving every factor. This avoids an exponential blow-up in possible configurations while still capturing control-relevant behavior. At the same time, interactions are not reward-dependent and thus do not require propagating difficult-to-achieve extrinsic rewards. By building up from easier-to-achieve interactions, more complex reward-optimizing behavior can be learned after an agent already has useful skills. Together, interactions provide rich, reachable subgoals which can be chained together to achieve more complex behaviors.

Our proposed Hierarchy of Interaction Skills (HIntS) algorithm identifies pairwise interactions in factored state spaces to learn a hierarchy of factor-controlling skills. HIntS adapts a causal test called Granger causality Granger (1969); Tank et al. (2021) with learned forward dynamics models to detect causal interactions and test factors for Granger-causal relationships. These interactions are then used to define the set of goals for goal-conditioned reinforcement learning, which learns skills that control the factors. Intuitively, this test determines if adding information about a new factor, which we refer to as the parent factor, is useful for prediction of the target factor (see Figure 2). If this is the case, then this suggests a causal link between the parent and target factors. By using statistical tests to discover the next factor to control, HIntS automatically discovers a hierarchy of skills starting with primitive actions. These skills control progressively more difficult-to-control factors, and the upper level can be used for reward optimization. In Breakout, HIntS discovers the intuitive sequence of skills described in Figure 1. We demonstrate that the HIntS algorithm not only learns the skills efficiently in the original version of the task but that the learned skills can transfer to different task variants—even those where the reward structure makes learning from scratch difficult. We show sample efficient learning and transfer in variations of the game Breakout and a Robot Pushing environment with high variation of obstacles, which are challenging domains for conventional RL agents.

Our work has three main contributions:

**1)** An unsupervised method of detecting interactions via an adapted Granger Causality criterion applied to learned forward dynamics models;

**2)** A hierarchical skill-learning algorithm (HIntS) driven by these discovered interactions; and

**3)** Empirical results demonstrating how HIntS, a proof-of-concept instantiation of interaction-guided goal-based HRL, can sample efficiently learn transferable, high-performance policies in domains where skills controlling pairwise factors can achieve high performance.

## 2 Related Work

Literature on skill learning is often split between learning reward-free (task-agnostic) skills and reward-based (task-specific) ones. Reward-free skills often use values derived from state visitation, such as information-theoretic skills Eysenbach et al. (2018), to make the skills cover as many states as possible while still being able to distinguish skills apart. Other reward-free skill learning uses tools such as the information bottleneck Kim et al. (2021) and dynamics Sharma et al. (2019). HIntS also learns skills based on state information. Rather than learning a skill to reach every state in an often intractable state space, we focus on controlling interaction-related factors. Interactions have some passing similarity to curiosity Burda et al. (2018); Savinov et al. (2018) and surprise-based reward-free exploration methods Berseth et al. (2019) in how it uses model discrepancy to set goals. HIntS introduces Granger-causal based interaction detection to HRL.

Another line of HRL methods backpropagate information from extrinsic rewards or goals Barto & Mahadevan (2003). Hierarchical Actor Critic Levy et al. (2017b;a) uses goal-based rewards in locomotion tasks and bears similarities to HIntS. Unlike HIntS, it uses complete states as goals and does not scale to complex object manipulation when there is a combinatorial explosion of configurations. Other hindsight goal-based hierarchical methods attempt to limit the difficulty of propagating information from the true reward signal through distance metrics Ren et al. (2019) or imitation learning Gupta et al. (2019). Alternatively, methods may learn skills end to end through neural architectures Bacon et al. (2017), option switching (termination) cost Harb et al. (2018), or variational inference Haarnoja et al. (2018a), though these struggle with the same sparse, long horizon reward issues. HIntS is similar to causal dynamics learning Wang et al. (2022); Seitzer et al. (2021), though these methods are non-hierarchical and focus on general causal relationships instead of interaction events. HIntS builds a skill chain similar to HyPE Chuck et al. (2020), which uses strong inductive biases with changepoint-based interactions. HIntS uses fewer biases using a learned interaction detector and goal-based RL. Altogether, HIntS offers a new way to define the skills for HRL in factored state spaces using interactions.

## 3 Background

### 3.1 Factored Markov Decision Processes

A Markov Decision Process (MDP), is described by a tuple $(\mathcal{X}, T, R, A_{\text{prim}}, \gamma)$. $\mathbf{x} \in \mathcal{X}$ is the state, $x_\circ \in \mathcal{X}_\circ$ is the initial state distribution, $a_{\text{prim}} \in A_{\text{prim}}$ are the primitive actions and $T(\mathbf{x}', a_{\text{prim}}, \mathbf{x}) \to \mathbb{R}$ is the transition function, a probability density function where $\mathbf{x}'$ is the next state. $R(\mathbf{x}, a_{\text{prim}}, \mathbf{x}') \to \mathbb{R}$ is the reward function, and $\gamma$ is the discount factor. The factored MDP (FMDP) formulation Boutilier et al. (1999); Sigaud & Buffet (2013) extends the state by assuming $\mathbf{x}$ can be factored: $\mathbf{x} \coloneqq \{x_1, \ldots, x_n\}$ where $\mathcal{X} \coloneqq \{\mathcal{X}_1 \times \ldots \times \mathcal{X}_n\}$. $\pi(\mathbf{x}, a_{\text{prim}}) \to \mathbb{R}$ is the policy, which is a probability distribution over actions conditioned on the state. In an MDP, the goal is to maximize expected return: $\max_\pi E_{\tau \sim \mathcal{D}(\pi)}[H]$, with $H \coloneqq \sum_\tau \gamma^t R(\mathbf{x}^{(t)}, a_{\text{prim}}^{(t)}, \mathbf{x}^{(t+1)})$, where $\tau$ is a trajectory $(\mathbf{x}^{(0)}, a_{\text{prim}}^{(0)}, \ldots, \mathbf{x}^{(\text{T})})$ and $\mathcal{D}(\pi)$ is the distribution of states induced by $\pi$. In the multitask setting, we assume a variety of tasks each described by $R_i \in \mathcal{R}$, and $\mathcal{X}_c irc^i \in \mathcal{X}_\circ$, or pairs of rewards and initial state distributions. We augment the state factors with non-state factors, such as $x_{\text{prim}}$ a factor describing primitive actions to discover interactions between primitive actions and state factors in the domain.

### 3.2 Options

A semi-MDP (sMDP) Sutton et al. (1999) augments the classic MDP with temporally extended actions called skills or options, defined by the tuple $\omega \coloneqq (I, \pi_\omega, \phi)$. $\Omega$ is the action space of the sMDP, and includes both primitive actions and/or other skills. $I \subset \mathcal{X}$ is the initiation set where a skill can be started. HIntS uses the common assumption that the initiation set for a skill covers the entire state space, though future work

could revisit this assumption. Goal-based skills parameterize the option policy and termination function by goals $c_\omega \in \mathcal{C}_\omega$, where the termination function ends a skill and typically takes the form $\phi(\mathbf{x}) = \|c_\omega - \mathbf{x}\| < \epsilon$, and the policy takes the form $\pi(\mathbf{x}, a_{\text{prim}}, c_\omega)$. Goal-based Hierarchical RL combines levels of skills Konidaris & Barto (2009) so that the goals of the parent in a chain (call it $\omega_a$) is the action space of its child $\omega_b$, with $\pi_b(\mathbf{x}, c_a, c_b) : \mathcal{X} \times \mathcal{C}_a \times \mathcal{C}_b \to \mathbb{R}$. This chaining can continue with arbitrary levels, *i.e.*, skills $c, d, \ldots N$, where $N$ is the depth of the hierarchy.

# 4 Hierarchy of Interaction Skills

This work, introduces an new interaction signal based on Granger-causal tests between factors in an FMDP, which captures interactions between factors as described in Figure 2. It then introduces an unsupervised HRL algorithm named the Hierarchy of Interaction Skills (HIntS) which uses sampled data to learn an interaction detector from source factor to target factor and skills to control the target factors. It then chains these skills to control new target factors, as illustrated in Figure 3.

HIntS uses factorized state so that $x_a \in \mathbf{x}$ is the state of factor $a$. $x_a$ is a fixed length sub-vector of $\mathbf{x}$ with $m_a$ features. We assume the factorization is given. The factorization problem is actively investigated in vision Voigtlaender et al. (2019) and robotics Lin et al. (2020) but is not the focus of this work. At one level of the hierarchy, HIntS detects the event when a source factor $a$, for which we have already learned skills, interacts with a target factor $b$.

In this section, we adapt Granger causality in the context of trajectories generated from a Markov Decision Process to define the interaction detector. Next, we describe how the interaction detector is used to learn a hierarchy of skills. HIntS iteratively learns interaction detectors and corresponding skills for the target factors one at a time.

## 4.1 Granger Causality

The Granger causal test Granger (1969) is a hypothesis test to determine if one signal $a$ with time series $x_a^0, \ldots x_a^T$ is useful for predicting a second signal $b$, $x_b^0, \ldots x_b^T$. The test follows first from the null hypothesis, which is the affine autoregressive (self-predicting) model of $b$:

$$\hat{x_b}^t = \beta + \left[\sum_{i=1}^{w} B^i x_b^{t-i}\right] + \epsilon_b \tag{1}$$

In this definition, $\beta$ and $B$ are learned by affine regression to best fit the data, $w$ is the history window, and $\epsilon_b$ is the noise. Then, the hypothesized relationship with signal $a$ is described by the pair-regressive distribution:

$$\tilde{x_b}^t = \alpha + \left[\sum_{i=1}^{w} B^i x_b^{t-i} + A^i x_a^{t-i}\right] + \epsilon_a \tag{2}$$

Signal $a$ is said to Granger-cause (G-cause) signal $b$ if the regression in Equation equation 2, $\tilde{x_b}^t$ provides a statistically significant improvement over the autoregressive distribution in Equation equation 1.

Granger causality is not equivalent to general causality Pearl (2009). However, this test proves powerful for detecting interactions, which is outside the scope of general causality and relates to the notion of actual or token causality Halpern (2016). While general causality distinguishes that the distribution of one factor has a causal effect on the distribution of another factor, actual causality describes the particular states where one factor causes another factor to take a particular value. We will discuss this relationship further in Section 4.3. Furthermore, in the context of stationary causal dynamics in fully observable MDPs, the Markov property makes it such that Granger causal tests can correspond to causal discovery when the state of the source factor is decorrelated from other causal parents of the target factor, *e.g.*, when it is assigned randomly by a state-controlling policy. See Appendix Section D for details.

## 4.2 Granger Causal Factor Tests

The relationship between source factor $a$ and target factor $b$ in an FMDP can be described using Granger causality with some adaptations to the linear regressive model. First, transition dynamics in FMDPs are not always affine, which requires modeling with a function approximator, as in recent work Tank et al. (2021). In this work, we use a neural conditional Gaussian model to capture both the prediction and the noise. Second, applying the Markov property allows us to collapse the history window to just the last state: $w = 1$. Combined, we describe the autoregressive and pair-regressive distributions with:

$$f(x_b)[x_b'] := P(x_b'|x_b) : \mathcal{X}_b \times \mathcal{X}_b \to \mathbb{R} \tag{3}$$

$$g(x_a, x_b)[x_b'] := P(x_b'|x_a, x_b) : \mathcal{X}_a \times \mathcal{X}_b \times \mathcal{X}_b \to \mathbb{R}, \tag{4}$$

where $x_b'$ describes the next target state after $x_b$, $f, g$ are functions mapping their inputs to a probability distribution over the next state: $P(\cdot|x_b)$, and $f(x_b)[x_b']$ and $g(x_a, x_b)[x_b']$ describe the probability density of the observed state $x_b'$ under the conditional distribution induced by $f$ and $g$, respectively. These models can be learned from data by maximizing the expected log-likelihood of the dataset $D$:

$$\max_{\theta} E_{(x_b, x_b') \sim D} \left[ -\log f^\theta(x_b)[x_b'] \right] \tag{5}$$

$$\max_{\phi} E_{(x_a, x_b, x_b') \sim D} \left[ -\log g^\phi(x_a, x_b)[x_b'] \right]. \tag{6}$$

We say a source factor $a$ Markov Granger-causes (MG-causes) the transition dynamics of a target $b$ if $g^\phi(x_a, x_b)[x_b']$ shows a statistically significant improvement in predictive power over $f^\theta(x_a)[x_b']$. Notice $a$ MG-causing $b$ indicates that $a$ is a useful predictor for the transition $P(x_b'|\cdot)$ compared to the autoregressive distribution. We test this with the MG-causal score $\mathrm{Sc}_{MG}$ for dataset $D$, where $L(g^\phi)$ abbreviates the log-likelihood of $x_b'$ expressed by the learned active model: $g^\theta(x_a, x_b)[x_b']$, and $L(f^\theta)$ the passive log-likelihood $f^\theta(x_b)[x_b']$:

$$\mathrm{Sc}_{MG}(D, g^\phi, f^\theta) := E_{x_a, x_b, x_b' \sim D} \log L(g^\phi) - \log L(f^\theta). \tag{7}$$

We call $g$ the active model, and $f$ the passive model, since the source factor $a$ is always drawn from either the primitive actions or a factor controlled with previously learned skills. This test can determine which target factors to learn new skills to control, though HIntS uses a modified version described in Equation equation 9 in the next section.

## 4.3 Detecting Interactions

While an MG-causal test describes whether source $a$ could be useful when predicting target $b$ in *general*, it does not detect the event where $a$ interacts with $b$ at a particular state $\mathbf{x}$ with a *particular* $(x_a, x_b, x_b')$. This distinction distinguishes general causality (the former) from actual causality (the latter), which is useful in many domains where an MG-causal source and target factors interact in only a few states. For example, in Breakout, while paddle to the ball is MG-causal, the paddle only interacts with the ball when the ball bounces off the paddle. In a robot block-pushing domain, while the gripper to block is MG-causal, the gripper only interacts when it is directly pushing the block. We are often particularly interested in those particular states where the ball bounces off of the paddle, and the gripper pushes the block. In this work, "interaction" is directional from source to target.

The interaction detector $I$ uses the active model $g^\phi$ and the passive model $f^\theta$ with the state transition $x_a, x_b, x_b'$ to detect interactions. Again, we use $L(\cdot)$ to denote the log-likelihood of the function evaluated at $x_b'$:

$$I(x_a, x_b, x_b') := \left( L(g^\phi) > 1 - \epsilon_{ACT} \right) \wedge \left( L(f^\theta) < \epsilon_{PAS} \right) \tag{8}$$

$\epsilon_{ACT}$ and $\epsilon_{PAS}$ are environment stochasticity hyperparameters. The first term describes when $g^\phi$ can predict the next state with high accuracy (probability higher than $1 - \epsilon_{ACT}$), and the second term describes when $f^\theta$ predicts with low accuracy (probability lower than $\epsilon_{PAS}$). The intuition for why this detects states where interactions between factors $a$ and $b$ occur is that states with high passive error $\log f(x_a)[x_b'] < \epsilon_{PAS}$ are

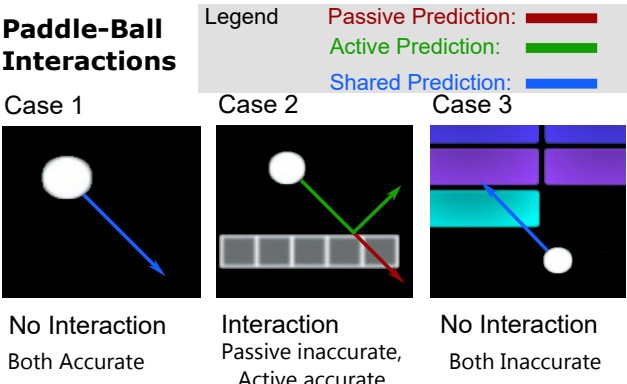

Figure 2: An illustration of the HIntS interactions. Case 1 is where both the passive and active model predict accurately, indicating that the active information is unnecessary and there is no interaction. Case 2 is where active information is needed to predict accurately, indicating there is an interaction. Case 3 is where the active information does not help predict accurately, indicating there is no paddle-ball interaction (though there may be a different i.e. ball-block interaction).

likely to have interaction *except* (1) when state transitions are highly stochastic, or (2) some other factor $c$ is interacting with $b$. However, these cases should elicit a high prediction error in $g^\phi$, which is agnostic to $x_c$. In the appendix, Figure 2 for a visual description in Breakout.

To search for interaction relationships between pairs of objects, we adapt the MG-score by scaling it with the interactions, giving the interaction score $\text{Sc}_I$, where $I(x_a, x_b, x_b')$ is abbreviated with $I(\cdot)$:

$$\text{Sc}_I(D, I, g^\phi, f^\theta) := E_{x_a, x_b, x_b' \sim D} I(\cdot) \left( \log L(g^\phi) - \log L(f^\theta) \right) \tag{9}$$

This score is more robust to the frequency of interactions, which might be extremely rare. In Breakout, for example, paddle-to-ball interactions only occur once every thousand time steps when taking random actions, so summary statistics can fail to efficiently capture this relationship.

### 4.4 Using Interactions in HRL

We now utilize the interaction detector in HRL. First, we use a factor that we have already learned skills for (or primitive actions) as the source $a$ and evaluate all uncontrolled factors (those without corresponding skills) with the interaction score (Equation 9). The highest score factor is used as the target factor. Then, we use the interaction detector to define goals in goal-based RL, the skill to control the target factor.

#### 4.4.1 Object Option Chains

The Hierarchy of Interaction Skills (HIntS) algorithm defines each layer of the hierarchy by goal-based control of one of the factors. Each option $\omega_i$ corresponds to one factor, *e.g.*, $\omega_{\text{Paddle}}$, or $\omega_{\text{Gripper}}$ with $\omega_1$ corresponding to primitive actions. HIntS then discovers a sequence $\omega_1, \ldots \omega_K$ which describes a goal-based skill to control each factor in the hierarchy using the skills $\omega_{i-1}$ it has already learned. Each skill is defined by a goal-based policy and termination condition where the action space of $\omega_i$ corresponds to the goal space of $\omega_{i-1}$, as described in the Background and illustrated in Figure 1. HIntS iteratively builds up from $\omega_1$, connecting a new factor $i+1$ to $i$ at iteration $i$ of HIntS. This iterative skill chaining is similar to that described in HyPE Chuck et al. (2020). To determine which $\omega_{i+1}$ to connect to $\omega_i$, we use the interaction score (Equation equation 9).

#### 4.4.2 Training Factor Control Skills

HIntS trains skills to control the target factor $b$ using goal-based RL, constructing the hierarchy described in Figure 3. While many goal-based RL methods rely on higher-level controllers to determine goals in the full state space, the high dimensionality of this space can often cause these methods to fail Levy et al. (2018). Instead, HIntS uses $I$ to filter the goal space in two ways. **1)** It only samples from the distribution

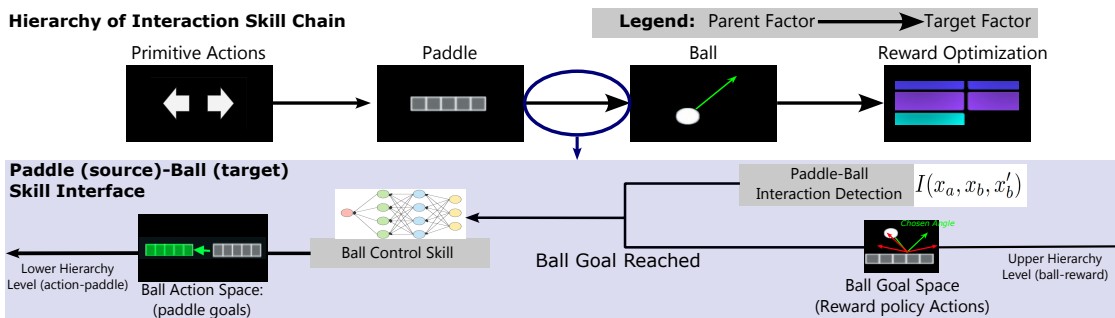

Figure 3: An illustration of the HIntS hierarchy. At each edge, a policy controls the target factor to reach an interaction goal, and takes interaction goals as temporally extended actions. The child of the primitive actions passes primitive actions as pseudogoals. **(lower)** Learning a single layer of the HinTS hierarchy, the pairwise interaction between the paddle and the ball. The interaction detector and ball skill use as inputs the paddle (parent) and ball (target) state and indicates if a goal is reached and a new temporally extended action is needed. The ball skill uses paddle goals as actions and receives ball bounces as goals from the reward-optimizing policy.

of observed post-interaction states $(x'_b)$ $\mathcal{C}_{b'} \subseteq \mathcal{X}_b$. **2)** It masks the feature space with $\eta_b$ to only include the components of the factor state that vary after an interaction, reducing dimensionality. We define $\eta_b$ as:

$$\eta_b[i] = \begin{cases} 1 & E_{x_a,x_b,x'_b \sim D}\left[I(\cdot)\left\|x'_b[i] - f^\theta(x_b)\right\|\right] > 1 - \epsilon_\eta \\ 0 & \text{otherwise,} \end{cases} \tag{10}$$

which is 1 when the feature $i$ of $x'_b$ varies significantly from the passive prediction at detected interactions. For examples of $\eta$-masks and post-interaction states, see Section 5. If the size of the masked post-interaction state set is sufficiently small: $\eta \cdot \mathcal{C}_{b'} < n_{\text{disc}}$, then we sample discretely, where $n_{\text{disc}}$ is a design choice. Using a fixed $\eta_b$, we define the termination function for our skill as:

$$\phi_b(x_a, x_b, x'_b, c_b) := \begin{cases} 0 & I(\cdot) \wedge \|\eta_b(x'_b - c_b)\| < \epsilon_c \\ -1 & \text{otherwise.} \end{cases} \tag{11}$$

We then learn a policy using reinforcement learning to reach these goal factors. HIntS uses only $x_a, x_b$ instead of all of $\mathbf{x}$ as input to the policy to accelerate learning by having the agent attend only to the minimum necessary aspects of the state. We train our goal-reaching skills using hindsight experience replay Andrychowicz et al. (2017), with Rainbow Hessel et al. (2018) for discrete action spaces, and SAC Haarnoja et al. (2018b) for continuous action spaces. A skill is considered reached when the goal-reaching reward remains unchanged after $N_{\text{complete}}$ training steps.

### 4.5 Building the Skill Chain

HIntS starts with primitive actions as the controllable parent object and takes random actions initially. At each iteration it: 1) learns active and passive models where $x_a$ is the state of the last controllable object. 2) Computes the interaction score for each target object as in equation 9. 3) Learns a goal based policy using the interaction model with the reward function described in Equation 11 for the factor with state $x_b$ with the highest interaction score. 4) Makes factor $b$ the controllable factor and restarts the loop.

Each step of the loop is executed autonomously, and the loop ends when none of the remaining factors have sufficient interaction score, $\text{Sc}_I < \text{Sc}_{\text{min}}$, where $\text{Sc}_{\text{min}}$ is a hyperparameter near 0, meaning that there are few to no detected interactions. Algorithm box 9 describes the algorithm.

### 4.6 Algorithm Overview

The algorithm is iterative, reassigning the parent factor $a$ to the target entity $b$ at the end of learning a skill for that factor, unless the loss does not converge (based on comparison with a random policy)

---

**Input:** FMDP Environment $E$
**Initialize** Hierarchy of Interaction Skills $\vec{\omega}$ with a single option set $\omega_{\text{prim}}$. Assign $a = $ primitive actions
**Data** $\mathcal{D} = \mathcal{D} \bigcup \text{random policy data}$
**repeat**

    **Interaction Detector**: Learn the passive and active models $f^\theta, g^\phi$, to get the interaction detector $I(x_a, x_b)$, as described in Sections 4.2, 4.3, for each factor as $x_b$.

    **Interaction Test**: Evaluate the performance of the passive model against the active model at interaction states using the interaction test in Equation 9 to determine a candidate target object $b$.

    **Option Learning**: Learn the interaction skills using hindsight goal-based RL with the goal function defined in Equation 11 if a candidate $b$ was found in the interaction test.

    **Update** $\omega_b$ to $\vec{\omega}$ if the option can reach random interaction goals. Assign $a = b$.
**until** HIntS no longer finds any interaction test pairs

---

## 5 Experiments

We systematically evaluate HIntS in two domains: 1) an adapted version of the common Atari baseline Breakout Bellemare et al. (2013) (specific differences in the Appendix Section A.1) and 2) a simulated Robot Pushing domain in robosuite Zhu et al. (2020) with randomly generated negative reward regions (Figure 1, also details in Appendix A.2). We compare HIntS against a non-hierarchical baseline that attempts to learn each task from scratch, a fine-tuning-based transfer with a 5m step pretrained neural model, and an adapted version of Hindsight Actor Critic (HAC) Levy et al. (2017b), a model based causal RL algorithm Causal Dynamics Learning (CDL) Wang et al. (2022), a curiosity and count-based exploration method, Rewarding Impact-Driven Exploration (RIDE) Raileanu & Rocktäschel (2020) and a hierarchical option chain algorithm Hypothesis Proposal and Evaluation (HyPE) Chuck et al. (2020). Our results show that HIntS is not only more sample efficient when solving the base task, is able to achieve higher overall performance, and also learns skills that generalize well to a variety of in-domain tasks in Breakout.

Details about the skill learning procedure, including the number of time steps used for learning each level of the hierarchical chain, the interaction masks and interaction rates are found in Appendix G for Breakout, Appendix H for Robot pushing and Table 4. In this section, we discuss the performance comparison with baselines. This learning procedure does not require human intervention—it progresses automatically based on the performance of the interaction score and the goal-reaching policies.

We represent all the policies, including the baselines, using a PointNet Qi et al. (2017) style architecture, which is order invariant and can handle an arbitrary number of factors. When there are only two factors, this is similar to a multi-layer perceptron. Details about architures are founds in Appendix F.

### 5.1 Overview of Baselines

In this section, we briefly describe the baselines and how they compare with the causal, interaction-dynamics and hierarchical components of HIntS. Even though Breakout and robot pushing have been evaluated before, not all baselines have been applied to these domains and we make some changes to ensure the baselines are on level performance. Furthermore, the Breakout variants and negative reward regions are novel tasks to demonstrate how reward-free Granger-causal skill discovery can provide significant performance benefits. The comparative baselines are listed below:

**HAC** Levy et al. (2017b) is a reward-based HRL method that uses goal-based learning to reach targeted parts of the state space, where hierarchical levels each have a horizon of $H$. This assesses a HIntS against a multilevel hierarchical method to illustrate how interaction-guided skills compares against a hierarchical method without interactions. HAC out of the box is infeasible in both the Breakout and Robot pushing tasks because of the high dimensionality of the state spaces. As a result, we introduce knowledge about the domain by having the skills set factor states as goals. In Breakout, the lower-level skills use the ball as goals. In Robot pushing, they use the block position. We tried other configurations in Appendix I but these are the only ones with nontrivial behavior.

**CDL** Wang et al. (2022) is a state-of-the-art causal method that learns causal links in the factored dynamics and uses the causal graph for model-based reinforcement learning. This illustrates how causal reasoning without interaction can struggle in rare-interaction domains with many factors. It does not learn to detect interactions but constructs a sparse graph to represent the counterfactual dynamics between state factors. In general, CDL struggles in these domains because it must learn models over all the objects, of which there are many. In addition, the model ends up being unable to help RL because it fails to capture the highly infrequent interactions, even when it picks up the correct causal graph.

**HyPE** Chuck et al. (2020) is an HRL method that builds a skill chain like HIntS using interactions and causal reasoning. However, interactions are defined by physical heuristics like proximity and quasi-static dynamics. In addition, instead of learning goal-based skills, HyPE learns a discrete set of skills. Nonetheless, it is the closest comparison for HIntS since it learns a hierarchy based on interactions, and it is included to illustrate how the Granger-causal interaction models and goal-based hierarchies allow more powerful and expressive skill learning. HYPE is designed for pixel spaces, but we redesign it to use the x,y positions of the factors directly. We introduce two forms of HyPE: R-HyPE uses the RL algorithm Rainbow Hessel et al. (2018) to learn polices, while C-HyPE uses CMA-ES Hansen et al. (2003). When evaluating C-HyPE sample efficiency, we add together the cost of policy evaluation for the policies and graph the performance of the best policy after each update, following the methods used in HyPE.

**RIDE** Raileanu & Rocktäschel (2020): Rewarding impact-driven exploration: is an intrinsic reward method that combines model error with neural state counts to perform efficient exploration. This method helps to identify if the benefits of HIntS comes from providing exploration through skill learning, rather than directed behavior from the interaction signal. RIDE has been applied successfully to visual agent maze navigation settings, but these domains often lack the factored complexity of Breakout or Robot Pushing. Our implementation of RIDE utilizes the Pointnet based architectures for the forward model with neural hashing, as well as Rainbow and SAC baselines. Additional details and discussion of RIDE can be found in Appendix I.4.

**Vanilla RL** uses Rainbow Hessel et al. (2018) for discrete action spaces and soft actor-critic Haarnoja et al. (2018b) for continuous action spaces, as these are the most stable RL methods for achieving high rewards on new tasks. We transfer vanilla RL by pretraining on the source task and then fine-tuning on the target task.

We chose not to compare with diversity-based methods for Breakout and Robot pushing because these domains are quite challenging for these methods, which tend to be limited to low-dimensional domains like navigation or locomotion, where the dimensions are limited to movement coordinates and joint angles. Breakout and Robot Pushing both have state spaces that are quite large: Breakout has 104 objects, and Robot Pushing has 18. Reaching a large number of these states is often infeasible. Designing a suitable diversity-based method for this would require a significant reengineering of existing work.

## 5.2 Sample Efficiency

In this section, we describe how HIntS is able to improve sample efficiency by learning skills directed toward particular interactions. Specific learning details can be found in Appendix A.1 and A.2, and training curves are found in Figure 4.

In Breakout, HIntS is able to learn to achieve high performance $4x$ faster than most of the baselines. This comes from utilizing the goal-based paddle control to shorten the time horizon of the ball bouncing task, and shortening credit assignment to the bounce interaction. HIntS achieves high rewards in this domain while learning to bounce the ball since good breakout performance by simply learning this behavior. HIntS skills exceed this by learning to bounce the ball at particular angles.

HAC is able to learn only the changes in Section 5.1. Even then, it performs poorly—worse than the vanilla RL baseline. We hypothesize this is because it is difficult to apply credit assignments properly. HyPE performs comparably in sample efficiency. C-HyPE even outperforms HIntS, because the evolutionary search can very quickly identify the paddle-ball relationship—which supports the hypothesis that interactions are

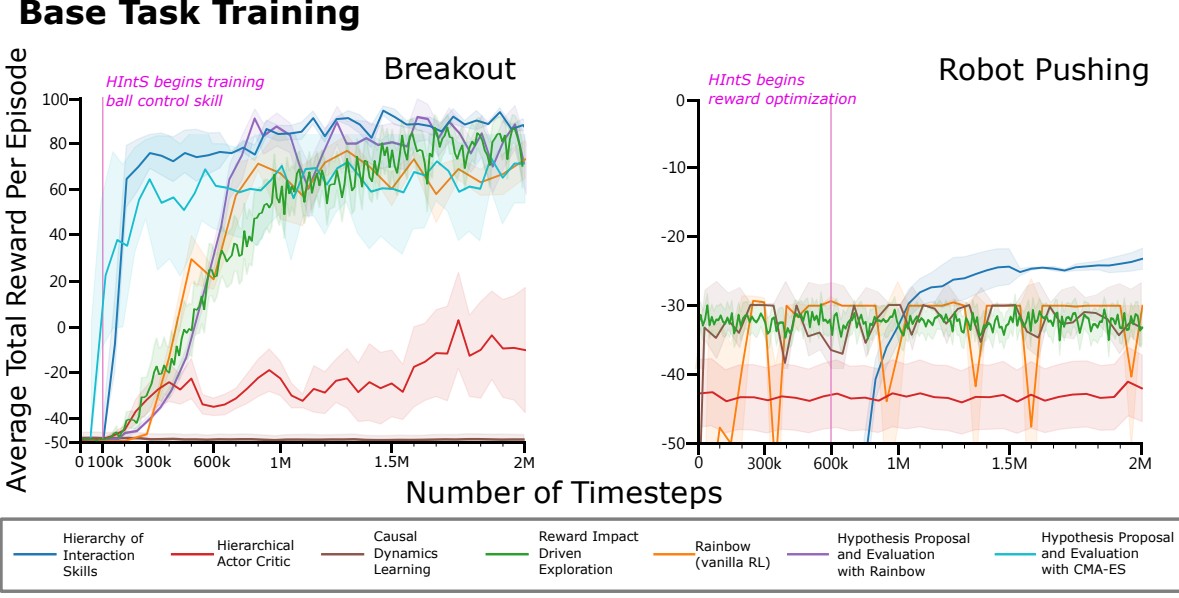

Figure 4: Each algorithm is evaluated over 10 runs with standard deviation shaded, y-scaled over the maximum. Training performance of HIntS (blue) against HAC (red), training rainbow/Soft Actor Critic from scratch (orange), CMA-ES-HyPE (cyan), Rainbow-HyPE (purple), CDL (brown) on Breakout (Top) and negative reward regions Robot Pushing (top). The vertical pink lines indicate when HIntS starts learning a policy that could achieve high reward (it has discovered the relevant Granger-causal dependence). In the pushing domain the return for not moving the block is $-30$, which is the score most of the algorithms converge at. The minimum total reward is $-600$—by spending every time step in a negative region, which is below the scale of the graph, and R-HyPE is in this area. Final evaluation of HIntS against the baselines after extended training is found in Table 1.

| | Base (5m) | HAC (5m) | R-HyPE (5m) | RIDE (5m) | HIntS (0.7m) |
|---|---|---|---|---|---|
| Break | $85.6 \pm 8.3$ | $74.3 \pm 8.2$ | $95.3 \pm 3$ | $92.5 \pm 8$ | $83 \pm 15$ |
| Push | $-30 \pm 0.01$ | $-43 \pm 24$ | $-90 \pm 19$ | $-31 \pm 2$ | $-21.6 \pm 4.6$ |

Table 1: Sample efficiency for Breakout and Robot Pushing with negative reward regions. Break is Breakout and Push is Robot Pushing. Note that $-30$ in Robot Pushing reflects the performance of a policy that never touches the block. HAC and Base are evaluated after 5m time steps of training, while HIntS gets equivalent or better performance 300k/700k for Breakout/Robot Pushing, respectively. The final performance after HIntS is trained for 1-2M time steps is shown in Table 5 in the Appendix.

a good guide for agent behavior. However, because of inaccuracies in identifying bounces and limitations of evolutionary methods, it does not learn to bounce the ball at desired angles.

HIntS sample efficiency results demonstrate how interaction-guided skills can learn complex behaviors more quickly by leveraging the interaction signal and shortening the time horizon. The comparatively similar performance of HyPE supports the efficacy of interactions since HyPE also uses interaction signals to achieve high performance. HIntS and other interaction-based methods are likely to perform well in domains where sparse interactions have a significant impact on performance, such as domains with sparse contact or distant objects.

## 5.3 Overall Performance

Many of the baselines can achieve near-optimal performance in Breakout, which is not the case in the Robot block pushing with negative reward regions task (see Appendix A.2 for details). In this task, none of the methods, even HIntS, achieves perfect performance. This is because the tabletop is densely populated with

| Algo | hard | neg | center |
|------|------|-----|--------|
| Base (5m) | $-7.1 \pm 0.8$ | $2.9 \pm 0.3$ | $-94.0 \pm 11.0$ |
| FT (5m) | $-6.1 \pm 0.4$ | $-7.0 \pm 19.0$ | $-61.0 \pm 9.0$ |
| HyPE $(< 2m)$ | $-5.1 \pm 0.49$ | $2.9 \pm 0.46$ | $-20 \pm 1.5$ |
| HIntS $(< 2m)$ | $-4.2 \pm 0.9$ | $3.6 \pm 0.3$ | $-46.0 \pm 4.0$ |

Table 2: Final evaluation of trained policies on Breakout variants. "FT" fine-tunes a model pretrained on the base task, and "Base" trains from scratch. The baselines are evaluated after 5m time steps, while HIntS achieves superior performance in 500k (hard, neg) and 2M (center). Figure 5 contains descriptions of the variants. For brevity, the remaining variant evaluations are in the Appendix in table 5.

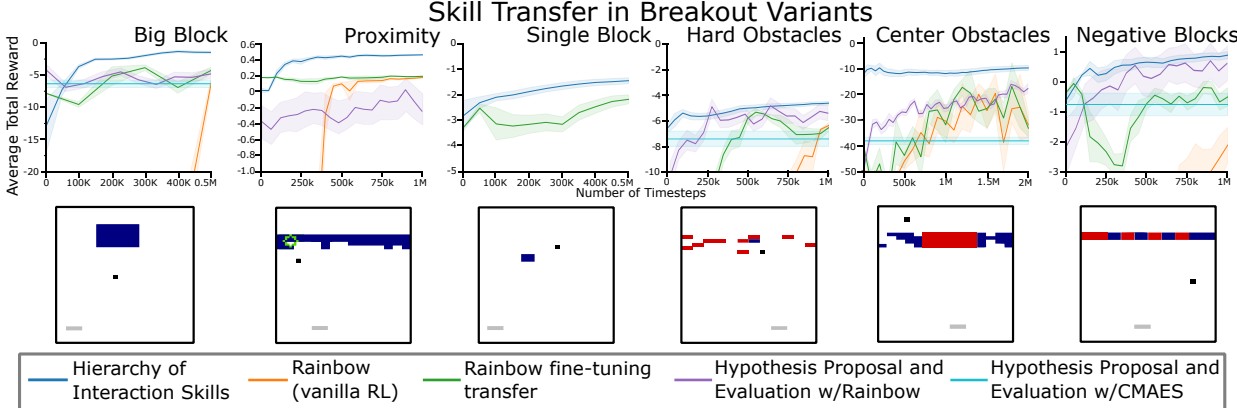

Figure 5: Skill transfer with HIntS (blue), training from scratch (orange), pretraining on default Breakout, and then transferring to the variant (green), and HyPE skills (C-HyPE in cyan, R-HyPE in purple). C-HyPE has nothing to optimize on transfer, so average performance is visualized. Below the performance curves, the variants are visualized with the red (light) blocks as negative reward/unbreakable blocks, and the blue (dark) blocks as blocks to target. Details in the Appendix I and Section 5.4.

negative reward regions in highly variable configurations between episodes, (see Figure 1). The task is also long horizon, taking as many as 300 time steps to complete, and reward shaping is difficult because pushing the block to the goal must be mediated by the arrangement of negative regions in between.

HIntS is able to discover block control through the interaction test and optimizes reward using block movements as actions. This results in a complex policy that pushes the block between the negative regions. None of the baselines can achieve this policy: CDL dynamics models fail to capture the rare dynamics of the gripper-block interaction. HAC and Vanilla RL use rewards, which causes the learned policies to never touch the block. HyPE policies can learn to move the block, but not the fine-grained control of pushing it to desired locations. See Figure 4 for details.

HIntS block pushing results demonstrate how Granger-causal interaction controlling skills break down an otherwise difficult task into one which is feasible for RL. Without using interactions, these skills are difficult to learn, as seen with policies learned with HAC. Interaction-based methods offer a possible direction for pretraining factorized control before optimizing reward.

## 5.4   Transfer

We demonstrate transfer in different variants of Breakout. These variants have complex reward structures that make them hard to learn from scratch such as a negative reward for bouncing the ball off the paddle, or for hitting certain blocks. These penalties make credit assignments difficult in a long horizon task.

**Single block**: A domain with a single, randomly located block, where the agent receives a $-1$ penalty for bouncing the ball, and the episode ends when the block is hit.

**Hard obstacles block**: The single block domain, but with 10 randomly initialized indestructible obstacle blocks.

**Big block**: A domain where a very large block can spawn in one of 4 locations, and the agent must hit the block in one bounce or incur a $-10$ reward and end of the episode.

**Negative blocks**: Ten blocks at the top give reward when struck. Half are randomly assigned to $+1$ reward, the other half $-1$ reward, and episodes are 5 block hits.

**Center obstacles**: The agent is penalized for bouncing the ball, and the center blocks are unbreakable, forcing the agent to learn to target the sides.

**Proximity**: a domain with a randomly selected target block the agent should hit as close as possible to for reward scaling between 1 and $-1$.

Because HIntS automatically learns skills to bounce the ball at a particular angle, these skills can transfer to learning these difficult Breakout variants. By reasoning in the space of ball bounces, the variants that have difficult reward structures have greatly reduced time horizons and simplified credit assignment.

CDL and HAC perform poorly on the original task, and it is unclear how to transfer their learned components. The fine-tuning strategy for transfer was the most effective baseline, as it transfers bouncing behavior to these new domains. However, performance rarely exceeds this behavior, and can even slightly decline. R-HyPE can transfer ball angle bouncing skills, but since the skills have many failure modes bouncing the ball, the overall performance on the variances is limited. C-HyPE does not learn to bounce the balls at different angles (it only learns a single option layer), so its performance is limited to the ball-bouncing policy it learns for Breakout. We took the average value of the C-HyPE policy.

These results, visualized in Figure 5, demonstrate how the skills learned with HIntS are agnostic to factors that are not part of the skill chain, i.e. the blocks. Like with robot pushing, these skills can exceed the performance of vanilla RL trained directly because they reduce the time horizon for complex reward assignments. In settings where at least some of the same objects and dynamics are present, GC interaction-based skills offer a transferable set of policies that can be used to achieve high performance even on difficult tasks.

## 6 Conclusion

This work presents a new approach to hierarchical RL in factored environments. The key idea is to use adapted Granger-causal interaction detectors to build a hierarchy of factor-controlling skills. We show this approach can solve tasks otherwise intractable for other RL methods, including other hierarchical methods, and learn transferable skills. The HIntS algorithm presents one practical method that performs well on a robotic pushing domain with negative reward regions and variants of the video game Breakout. Future work can address the limitations of HIntS. In particular, two of the primary limitations are the reliance on pairwise interaction assumptions and on the factorized state. Because HIntS learns a pairwise skill chain, in domains where large numbers of interactions are present, HIntS may learn a chain of skills unable to suitably solve any task in the domain. In addition, the skills are learned using a binary interaction signal, which does not differentiate between differents kinds of behavior *within* a pair of objects, such as differentiating pushing and picking. While these limitations are significant, HIntS demonstrates a first step in using interactions between factors. In addition, future work can investigate tree and graph-structured skill hierarchies, utilize human demonstrations to handle tasks like grasping, and further differentiate skills while relaxing sparsity assumptions, hyperparameter complexity, tuned stopping conditions and computational cost. Despite these limitations, HIntS shows evidence that controlling interactions in state factors is a promising approach for skill discovery.

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

# A  Environments

## A.1  Breakout

The Breakout domain contains as factors the actions, ball, paddle and blocks, shown in Figure 1. We describe the training process for HIntS iteration by iteration both to illustrate results and clarify the algorithm.

The Breakout domain is not the same as Atari Breakout. It contains 100 blocks, and the agent receives $+1$ reward whenever it strikes a block. An end of episode signal, and -10 reward is given whenever the agent lets the ball fall past the paddle. All other rewards for the task are 0. The ball moves at $\pm 1$ x velocity and $\pm 1, \pm 2$ y velocity, depending on how it is struck by the block. Like in Atari Breakout, the ball angle is determined by where on the paddle the ball strikes, with four different angles.

We use an adapted Breakout environment for three reasons. First, we want the domain to have stationary dynamics so that learning dynamics models is relevant. However, the Atari Breakout domain has the velocity of the ball change based on the number of bounces the agent has taken, where the bounces is a hidden variable. Rather than overcomplicate the learning procedure, we opted to use a domain where this feature was absent. Second, to investigate long horizon tasks we wanted the domain to have sufficiently long episodes, which we accomplished by slowing the speed of the ball. This is typically worked around in RL algorithms through frame skipping to improve efficiency, but we are actually interested in investigating the granular features of the Environment like the transition dynamics. Third, since we are working in a factored space we wanted a domain where we could have easy access to the factors, and related statistics like which objects are interacting with which others at a given time step. This is easier to accomplish by having a custom version rather than trying to extract that information from the RAM state of Atari Breakout.

## A.2  Robot Pushing

The robotic pushing domain uses as factors the actions, gripper, block, and negative reward "obstacles". The gripper is a 7-DOF simulated Panda arm generated in robosuite Zhu et al. (2020) with 3-DOF position control using Operational Space Control (OSC). The OSC controller generates joint torques to achieve the desired displacement of the end effector specified by the action. The objective of the domain is to push a $5\text{cm}^3$ block to a 5cm diameter circular goal location randomly placed in a 30cm×30cm spawn area. The area is broken into a 5×5 grid, where 15 of the 25 6cm×6cm grids are negative reward regions (NRR), which implies almost $300M$ possible configurations of the obstacles, block, and target, forcing the agent to learn a highly generalizable policy. If the block enters an NRR, or leaves the spawn area, the agent receives $-2$ reward. The negative regions are generated such that there will always exist an NRR-free path to the goal. The domain also has a constant $-0.1$ reward per timestep for goal reaching, with episodes of 300 time steps. An image of the domain is shown in Figure 1.

The negative reward region locations are generated such that there will always be a trajectory to the goal location, so the actual number of possible configurations is limited by this. However, this is still more than enough complexity for most problems, and in fact, this could be framed as a generalization in RL problem, since it is entirely possible in millions of timesteps that the agent has never seen the provided configuration. In other words, the agent must learn to actually reason about the negative reward regions, something that is really only possible with current RL algorithms if it reasons from the space of the block positions, and not from the gripper, hence our choice.

We would have liked to use a pick-and-place task, but these tasks become infeasibly difficult to perform from purely random actions, often requiring clever hacking of the data to get the gripper to grasp the block. This would undermine the causal nature of the tests since now the actions are chosen without being agnostic to the objects. However, we are looking for adaptations

# B  Details on Active and Passive Model Training for Interactions

The interaction model often requires data balancing in order to train. For one thing, the policy is the mechanism used to gather data, and when it is taking anything other than random actions, this can induce

a correlation between the policy and the behavior of the objects. To mitigate this, we only train the passive model on states where the actions are taken from a policy agnostic to $b$. In practice, this is often sufficient to train the passive model for good prediction on states where interactions do not occur, though we have found that using a heuristic to determine which states interactions are not happening, such as proximity, can greatly smoothen the learning process. This is because the model can often fixate on the states that it cannot predict, resulting in misprediction at states where it could otherwise perform well. This also helps to widen the gap between the passive and active model log-likelihoods, which makes $I$ more accurate

While perfect data and modeling give an $I$ that captures most interactions, interactions can vary from being extremely rare (the ball bouncing off of a randomly moving paddle), or extremely common (the actions affecting the paddle everywhere). In practice, this makes data balancing an issue when training the active model $g(x_a, x_b)$, where a network will end up struggling to predict the states where $x_a$ is useful because it is overwhelmed by the volume of passive states. Simply adding model complexity can cause the active model to memorize states which might be hard or involve an interaction with a different factor $c$, resulting in spurious interactions.

To combat this, HIntS uses two additional methods: first, we upweight states with high passive error (low log-likelihood) when training the active model. This causes the active model to favor predicting states where the passive model is already performing poorly. Second, we scale these data sampling weights with proximity to prevent the active model from overfitting to all states with high passive error, including ones where $a$ does not interact with $b$. This keeps the active model from memorizing any state. Future work could consider other, general strategies for controlling data imbalances.

These two data balancing methods can be combined in an unnormalized weight $w$ for any state that has the following passive error and proximity, with hyperparameter $\lambda$:

$$w_b = \begin{cases} \lambda + 1 & -\log f(x_b)[x_b] > 0 \wedge \|x_b - x_a\| < \epsilon_{\text{close}} \\ 1 & \text{otherwise} \end{cases} \tag{12}$$

We also propose a possible way to tune the active model with interactions, by weighting the loss of the active model by the interaction model. This has to be balanced from overfitting, however, as this can make the interaction model boost the active model. This tuning would make the forward loss with weighted dataset $D_w^b$:

$$\min_\phi L(g) := E_{x_a, x_b, x_b' \sim D_w^b}[\lambda_i \log \left(g^\phi(x_a, x_b)[x_b']\right)$$
$$+ (1 - \lambda_i)I(x_a, x_b, x_b') \log \left(g^\phi(x_a, x_b)[x_b']\right)] \tag{13}$$

Where $\lambda_i$ is the mixing parameter. However, combined with the proximity this can end up being too much boosting and result in overfitting in the active model.

We also tried training an interaction model, which is a function $I^\theta(x_a, x_b) : \mathcal{X}_a \times \mathcal{X}_b \to [0, 1]$, the probability of an interaction at a given state, and learns to predict the interaction detections $I(x_a, x_b, x_b')$. This could have the benefit of generalizing the characteristics of an interaction to states where $x_a'$ might be hard to predict. However, because we continuously train the active model with new data from the policy, this turns out not to perform well because it complicates the learning process.

## C   Object Centric Actions and State Augmentations

When using the goal space of the current option as actions, there two important design decisions: should the action space be continuous or discrete, and should the actions be in a fixed space, or relative to the state of the target object.

In the first case, while in general goal-based RL there is no significant choice between discrete and continuous action spaces, in cases where the goal space is small, such as the ball velocity, discrete spaces are preferable. This is because the space of states seen after an interaction $\mathcal{C}_{b'}$, after being further reduced by the controllable elements mask $\eta_b$, can be quite small. Small enough, in fact, that the historical values might number less

than 10. When this space is small, the locality that is present in normal state spaces might not be present, so selecting continuously could add bias to "nearby" states. Thus, if $\mathcal{C}_{b'} \cdot \eta_b <= 10$, or there are less than 10 seen interaction states, then we use discrete actions.

For the second case, continuous actions are generally better as object relative. This is because the relationship between objects is also almsot always based on their relative state. This is done by having the policy output actions $a_\pi$ between -1 and 1, and then remapping that into a relative action space, or $a_\pi \cdot d \cdot \eta + x_b$, where $d$ is the relative distance a single action can go, which is typically 0.2 of the overall state space (we normalize the state space).

## D   Causality and Object Interactions

While this work introduces a novel metric for identifying entity interactions, similar ideas in mechanisms like learning causal relationships Pearl (2009) have been incorporated in model-based and planning-based methods to explain dynamics Li et al. (2020); Wang et al. (2022). Schema networks Kansky et al. (2017) and Action Schema Networks Toyer et al. (2018) extend causal models to planning. Alternatively, variational methods such as Lin et al. (2020) learn to identify objects in a visual scene for planning Kossen et al. (2019), or reinforcement learning Veerapaneni et al. (2020). By contrast, HIntS limits the modeling burden by only capturing forward dynamics models good enough to identify interactions and using them for HRL, drawing ideas from empowerment Jung et al. (2011) and contingency Bellemare et al. (2012) to improve exploration. The model-disagreement technique used by HIntS closely mirrors neural methods for Granger Causality Granger (1969); Tank et al. (2021), though this is a novel application to HRL.

In this work we differ from general causality in two ways: first, we acknowledge that the Granger-causal test does not demonstrate causality, but rather is a predictive hypothesis test designed for time series. However, in the case of forward time ($\mathbf{x}'$ cannot cause something in $\mathbf{x}$ or any prior state), fully observable (there are no unobserved confounders) and Markov ($\mathbf{x}'$ depends only on $\mathbf{x}$) dynamics, our MG-causal test (Equation 9) this matches a causal discovery test under two conditions: 1) There is no shared information true cause $x_c$ of $x'_b$, 2) The selection of $do(x_b)$ is decorrelated from $x_a$.

In the first case, this describes the case where some factor $c$ confounds $a$ on $b$ that is, $P(x'_b|x_b, x_a, x_c) \neq P(x'_b|x_b, do(x_a), x_c)$. Note that because of the Markov assumption $x_c$ is NOT a common cause of $x_a, x'_b$. Instead, this occurs when factor $a$ shares some information with factor $c$, that is $P(x_c|x_a) \neq P(x_c)$, and $x_c$ is the true cause of the transition dynamics of $b$. In our case, because we choose the factor pair with the highest MG-causal test score, meaning that $x_a$ must be a better predictor of $x'_b$ than $x_c$, *despite* not being the true cause. As an example of this, imagine that in Breakout factor $a$ describes the paddle which rarely interacts with the ball, while factor $c$ is another paddle with noisy observation whose location is determined by factor $a$. In this case, where factor $b$ is the ball and we search for a $c \to b$ relation, then factor $a$ may confound this information because information about paddle $c$ bounces is subsumed in paddle $a$. This condition is rare but possible in many real-world or real-world-inspired domains, and it is a limitation of using pairwise scope relationships.

In the second case, if $do(a)$ is not selected randomly but based on $x_b$, and this difference is not accounted for, then the Granger-causal relationship is not captured. This is the case if the input data for $x_a$ has randomly assigned $x_a$, for example if the goals for the policy are sampled randomly and the policy has no dependency on $x_b$ (which we use in this work). An alternative way to correct this would be to correct values with the importance sampling weights $\frac{P(x_b|\pi_{\text{directed}})}{P(x_b|\pi_{\text{random}})}$, where $\pi_{\text{directed}}$ is the policy with imbalenced behavior.

Second, we describe interactions using similar language to actual causality because we are implicitly assuming that an interaction occurs when one object is the cause of one behavior in another object in a particular state. In actual causality, the causal relationship is not described in the general case (*could $a$ cause $b$*), but in the specific case between factors (did $a$ cause $b$ in particular state $\mathbf{x}$). The active and passive models also capture this, because the passive model is trying to capture the "what would have happened", and the active model is capturing "but for $\alpha$ in a particular state or relation to $\beta$."

This work in entity interactions is also related to the field of actual causality because we are implicitly assuming that an interaction occurs when one object is the cause of one behavior in another object in a particular state. The active and passive models also capture tis, because the passive model is trying to capture the "what would have happened", and the active model is capturing "but for $\alpha$ in a particular state or relation to $\beta$."

## E    Transferring Skills

When we say that we transfer the learned hierarchy from HIntS, this means that we take the final skill, whether it is that it hits the ball with high accuracy at one of four desired angles in Breakout, or moves the block to a specified location in robot pushing, and use this goal space as the action space for a high-level policy that takes in all of the environment states. Thus, both the action space for this new domain is temporally extended according to the length of time it takes to reach the desired goal, or a cutoff. These actions will still call lower level actions (paddle or gripper control), as necessary. In Breakout, we hypothesize that temporally extended skills on the order of ball bounces, which is around 70x less dense, allow for better credit assignment, causing performance improvement. We replaced the hierarchy with a known perfect ball-bouncing policy with slightly better results, supporting this idea. In robot pushing, we suspect that the benefit comes somewhat less from temporal extension, and more from the fact that the action space is much more likely to move the block (compared with primitive actions that move the gripper), which allows for better exploration.

As a sidenote, we could have learned a hierarchy up to controlling the blocks in breakout. However, this would have required added complications to the algorithm: first, it is not always possible to hit a block, either because there are other blocks in the way, or because the block simply does not exist. This would require us to either hard-code a sampler, or train one that learns these properties. Next, if we were to transfer this policy, it would then require a policy that chose a block location to hit, either discrete selection or a continuous location, which would put onus on the learner to figure out how to hit it. Finally, learning the block policy is sample inefficient, and is likely not to give much benefit, and we can stop learning at any level of the hierarchy. As it is, a block targeting policy is learnable, since the proximity variant essentially captures this.

## F    Network architectures

For this work we use a 1-d convolutional network with five layers: $128, 128, 128, 256, 1024$ which is then aggregated with max pooling and relu activations followed by a 256 linear layer before either rewards, critic, passive distribution output, active distribution output. In every case, the inputs are the $x_a$ and $x_b$ object(s), where we assume only a single $a$, though possibly many targets. In the final evaluation training, $a$ is the set of all factors that are not multiple, except for primitive actions in policy training. We then append $x_a$ to $x_b$, and treat each of these as a point. There are also well-known issues in Breakout that without a relative state between the paddle and the ball, the policy fails, so we augment the state with $x_a - x_b$ and $x_b - c$ when appropriate (the dimensions match). Combined, we call this architecture the Pair-net architecture (a Pointnet architecture for pairs of factors)

We can use the same architecture for all of the policies. Note that when there are not multiple instances of objects (i.e. cases without the blocks in Breakout or the Negative reward regions in Robot Pushing), the Pairnet architecture reduces to a simple multi-layer perceptron. For the sake of consistency, we used the same network for every choice, though this was overparameterized in some cases.

## G    Breakout Training

In this section we report the learning process for HIntS in Breakout. Note that the algorithm uses automatic cutoffs to decide which factors to learn a policy over using the interaction score in Equation 9 and comparing it against $\epsilon_{\text{edge}}$, a minimum cutoff weighted log-likelihood. In practice, we found that 5 worked for all edges in Breakout and Robot Pushing, though this can probably be automatically learned based on an estimate of

environment stochasticity. The training curves can be seen in Figure 6, and visualizations of the skill spaces in Figure 7.

### G.0.1   Paddle Skill

HIntS collects random samples (10000 sample intervals) until it can detect the action-paddle connection through the interaction score (Equation equation 9). Interaction tests between other factor pairs—action-ball and action-block, have low scores, with details of the comparative performance in the Appendix. The paddle interaction detector "detects" every state as an "interaction", since the actions control the paddle dynamics at every state. $\eta_{\text{paddle}}$ masks out all components except the x-coordinate of the paddle: $[0, 1, 0, 0]$ since it can only move horizontally, and $\eta_{\text{paddle}} \cdot \mathcal{C}_{\text{paddle}'}$ is the x-range of paddle locations. The paddle skill uses $x_{\text{paddle}}$ as input and learns to perfectly move the paddle to a randomly chosen target x-coordinate in roughly 10k time steps. Paddle training ends when the success rate over 100 updates no longer decreases.

### G.0.2   Ball Skill

HIntS continues to gather samples in $10k$ increments using the paddle policy. For every $10k$ new samples, the interaction detectors between the paddle and other factors are updated, and if the performance exceeds $\epsilon_{\text{edge}}$, a skill for that factor is automatically learned. HIntS discovers the paddle-ball interaction with the interaction score reported in Table 4 after 80k additional samples. It takes this many time steps because of the infrequency of ball interactions. Using $I_{\text{ball}}$, HIntS discovers $\mathcal{C}_{\text{ball}'} \cdot \eta_{\text{ball}}$ with four velocities $[-1, -1]$, $[-2, -1]$, $[-2, 1]$, $[-1, 1]$, the four directions the ball can be struck in Breakout, with $\eta_{\text{ball}} = [0, 0, 1, 1]$. Since $|\eta_{\text{ball}}\mathcal{C}_{\text{ball}'}| < n_{\text{disc}}$, we sample discretely from this set.

The discovered interaction implies a skill trained with hindsight to hit the ball at the randomly sampled angle. Notice that this problem subsumes the one of just playing Breakout, which just requires bouncing the ball. The resulting skill run with random ball velocities plays Breakout well after $< 100k$ time steps as seen in Figure 4 and Table 1, though at that point it only has $\sim 30\%$ accuracy at hitting the ball at the desired angle. Since the success rate has not converged over 100 updates, training continues for $1m$ time steps to achieve 99.5% accuracy for striking the desired angle. At this point, HIntS terminates, though future skills such as block targeting were trained as a variant. Since blocks are separate factors, their individual interaction scores are low due to data infrequency. A class-based extension would allow HIntS to handle this.

## H   Robot Pushing with NRR Training

The negative reward regions pushing task is difficult, and all these baselines fail as seen in Table 1. Even intrinsic reward methods, like RIDE, struggle in this domain, possibly because the intrinsic reward signal is washed out before the agent can learn to manipulate the block, and and sufficient intrinsic reward can be acquired simply manipulating the gripper Only HyPE has non-trivial reward, because the options it learns will force it to move the block. However, the skills navigate it to negative reward regions or out of bounds most of the time instead of the goal, resulting in lower trajectory reward than just not moving. In most cases, the agent is highly disincentivized to push the block at all since the likelihood of reaching the goal is low, but the likelihood of entering a negative region is high. However, using a smaller penalty would cause directly pushing to the goal to be optimal—it takes several time steps to push the block around an obstacle. We tried a modified baseline, which pretrains the agent to push the block to the target, and then fine-tunes the policy to avoid obstacles, but that baseline policy still ends up quickly regressing to inaction. Only HIntS achieves non-trivial success, with an average performance of $-21.6$. Importantly, the agent can reach the goal consistently, though it occasionally incurs penalty for slipping into the negative reward zones because the block skill is agnostic to the NRR. Future work could consider training the policies end to end so that NRR information can be used when learning lower level skills.

We illustrate the HIntS training curves in the Robot Pushing domain in Figure 8, and visualize the goal spaces in Figure 9. We describe the learning procedure here: HIntS first gathers 10k random actions according to the same strategy described in Breakout training—it automatically discovers edges using the Interaction

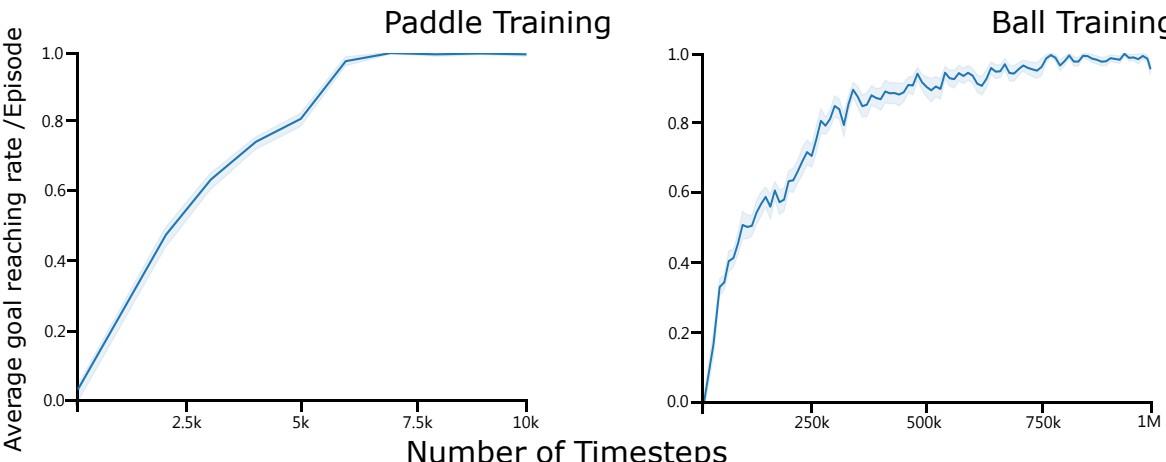

Figure 6: This graph shows the training curves for HIntS learning the Breakout skills, where a rate of 1.0 means that the goal is reached on 100% of episodes. We averaged performance over 10 runs. Note the difference in scale between the paddle skill learning (10k time steps), and ball training (10M time steps). Training terminates automatically when performance converges (train goal reaching rate does not increase).

.

test. In this domain, the majority of the benefit arises from based on the amount needed to discover useful correlation with action. It enumerates edges between the primitive actions and the different objects in the domain and learns object models according to the method described in Section 4.2, 4.3.

The action-gripper model passes the interaction test with the active model having a weighted performance of 4cm better than the passive model, while the spurious action-block edge fails with less than 0.1cm difference, as enumerated in Table 4. Even though designed for sparse interactions, the model is able to pick up that the action affects the gripper at almost every time step. The controllable set operation from Section 4.3 returns a mask of $[1, 1, 1]$ since all the gripper components are sensitive to actions. Since the parameter set $c_{\text{gripper}}$ is $> 10$, the goal state is sampled continuously not discretely.

The gripper policy is then trained to reach randomly sampled gripper positions. The gripper model takes as state the state of the gripper and the last action. The gripper policy converges to 100% accuracy at moving within 0.5cm of target positions within 50k time steps. However, again following the $10k$ sampling procedure for gripper-block interaction, we continue to gather gripper movement behavior until gathering a dataset of 100k time steps.

HIntS then samples the edge between the gripper and the block, appended with action information as a "velocity" for the gripper. With the combined input state, the forward model passes the interaction test with about 1.5cm better-weighted performance. This difference is smaller because block interactions from random actions are rare, and can be low magnitude—the robot only nudges the block. The mask trained over the controllable area is $[1, 1, 0]$ since the block height does not change (the gripper cannot grasp). The block policy learns to move the block to within 1cm of accuracy, and the block-pushing skill converges within 500k time steps.

Learning the final policy with the HIntS skills takes around $700k$ time steps. This is because the goal-based movement of the block allows the agent to navigate carefully around objects. However, the task is sufficiently difficult that even with that, the agent still plateaus in performance even though a human using the learned options could get a better reward. This is probably because there are many local minima. This is where a model that predicts block movement could be incredibly useful for learning because it would allow imagining multiple trajectories without getting collapsed into a single low-reward one.

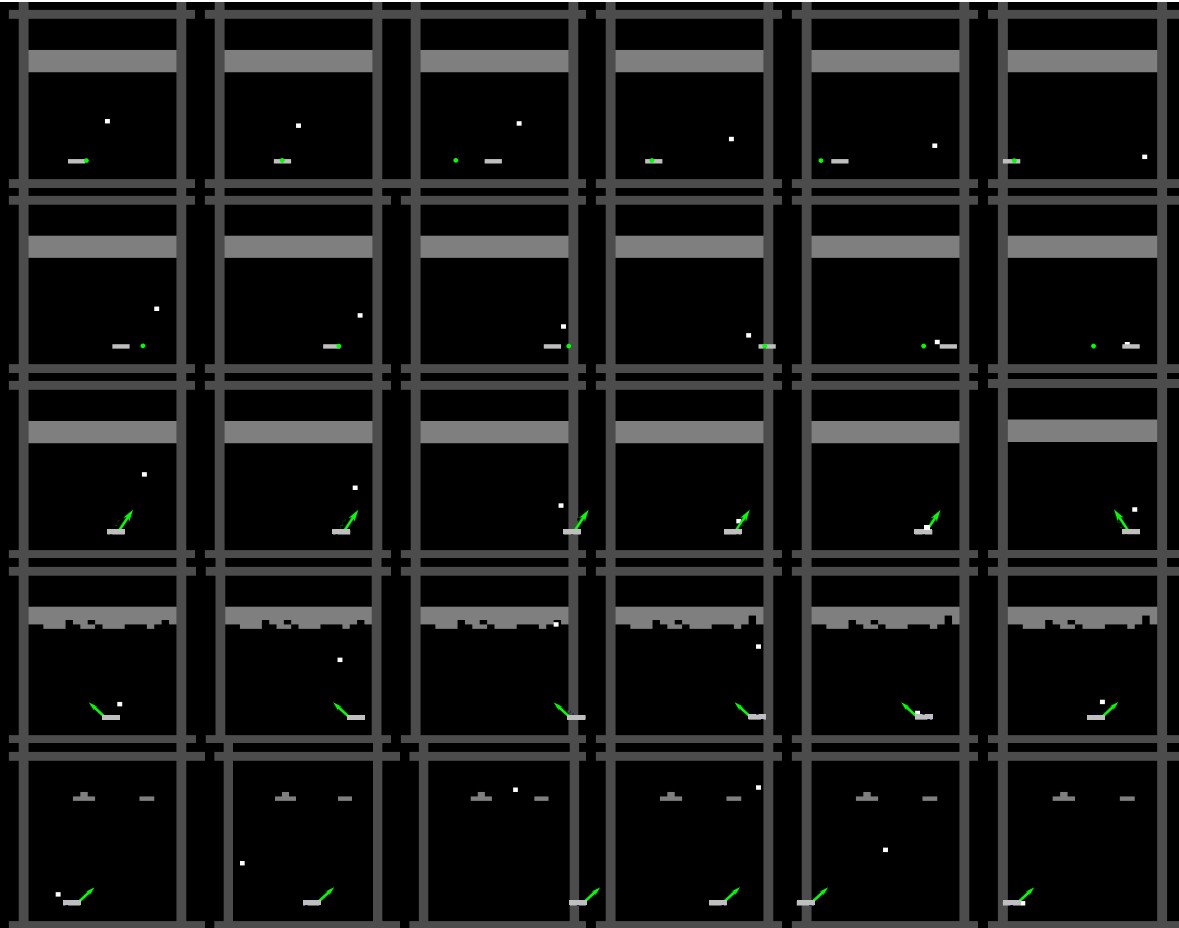

Figure 7: Visualization of the Breakout skills. The first two rows illustrate the block, where the green dot is the desired goal position. The last three rows show the target ball angles, with each row illustrating how the agent manipulates the paddle to produce the desired post-interaction angle. There are four possible angles.
.

# I   Additional Baseline Details

## I.1   Hypothesis Proposal and Evaluation (HyPE)

The HyPE algorithm relies on discrete action spaces which requires changing the robopushing environment by constructing a discrete action space for HyPE to use. This space consists of actions in the cardinal directions x,y,z, where the agent moves 1/10 of the workspace for each action. This was tested by having a human perform demonstrations with this workspace. HyPE also performs much better in breakout, where the quasi-static assumptions capture the relationship between the paddle and ball especially well since changepoints capture the instantaneous motion easily. However, in the robot-pushing domain it struggles because the block movement can vary in magnitude, but the nature of the actions is such that any movement will get assigned to an action completion. As a result, learning the block-pushing options are only somewhat effective, able to move the block in some regions of space, but getting stuck in others. HyPE is the only baseline that actually performs meaningful learning on the base task, starting from $-400$ reward to only $-90$. Ironically, it also has the lowest performance because the other baselines just never touch the block.

HyPE has two varieties, one where the agent is trained using Covariance matrix adaptation evolution strategy (CMA-ES) Hansen et al. (2003) (C-HyPE), and another where it is trained with Rainbow Hessel et al. (2018) (R-HyPE). In the CMA-ES case, while the evolutionary algorithm is good at picking up easy relationships

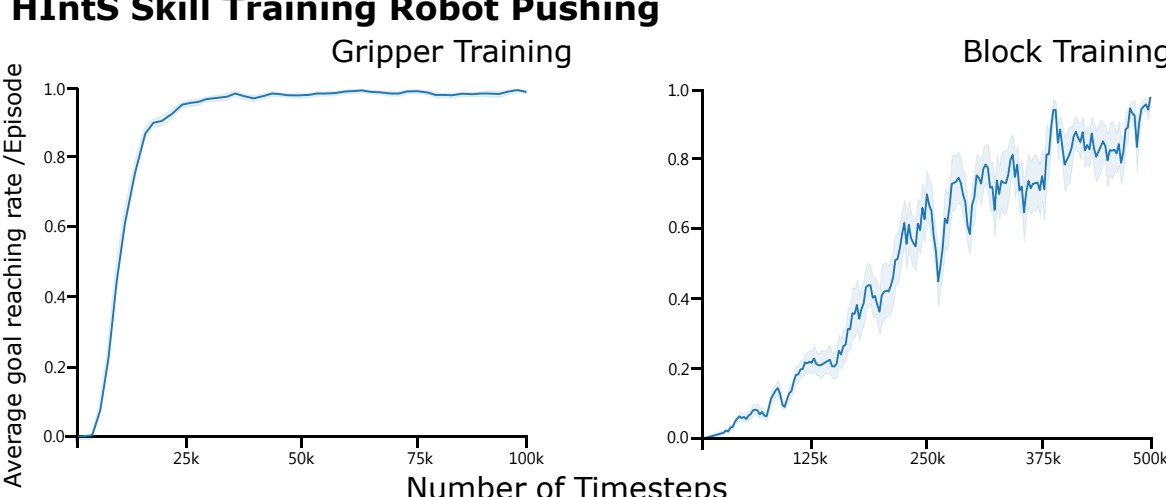

Figure 8: This graph shows the training curves of HIntS learning the Robot pushing skills, where 1.0 means a goal is reached (interaction and within epsilon of desired state) on 100% of episodes. We averaged performance over 10 runs. Note the difference in scale between the gripper skill learning (100k time steps), and block training (500k time steps). Training terminates automatically when performance converges (train goal reaching rate does not increase).

.

like the ball to the paddle, it cannot actually learn multiple policies to hit the ball at each desired angle, and only trains to create a bounce changepoint (the next layer that would be used for transfer has only one action, which is why C-HyPE is not trained at transfer). Thus, the CMAES version performs well at the main task and poorly at the overall task. On the other hand, because HyPE learns a separate policy for each ball angle, it takes much longer to learn. In fact, these policies struggle to keep the ball from dropping (letting the ball go past the paddle), which is the main reason for the poor transfer performance when compared with even the pretrained-fine tuned baseline.

C-HyPE is trained with a 128-activation hidden layer multi-layer perceptron, because it needs an $n^2$ computation in the number of parameters, so using the larger PointNet architecture is infeasible. This is another reason why it struggles to learn complex policies.

By comparison, HyPE and HIntS learn similar object-option chains, but because the interaction detector is more robust and the goal-based option is more general, HIntS is able to perform when HyPE cannot. HyPE also makes stronger assumptions about the environment, including the quasi-static assumption and discrete actions. The reliance on discrete modes for high level skills also limits HyPE.

### I.2 Causal Dyanamics Learning

We ran CDL in every domain (Breakout and Robot Pushing) and all the Breakout variants. It failed in every domain. We hypothesize this is because when the data that it gathers is not particularly meaningful, which is true of random actions in both Breakout and Robot Pushing, it struggles to construct a model which is useful to the agent. Without temporal abstraction, the policies struggle, and this lack of temporal extension is exacerbated by a model that makes inaccurate predictions about key predictions (object interactions). Even with a good model of the system, both these domains have very long time horizons, making them very challenging for on-policy algorithms like PPO (which CDL uses for the model-based component). Random shooting only helps for certain special states (before an interaction), but detecting those states can be challenging, and the rest of the time, it gives back sparse signals. On top of the fact that the rewards are sparse, this is why the domains end up being challenging for CDL. We include a table of performance for CDL in Table 5

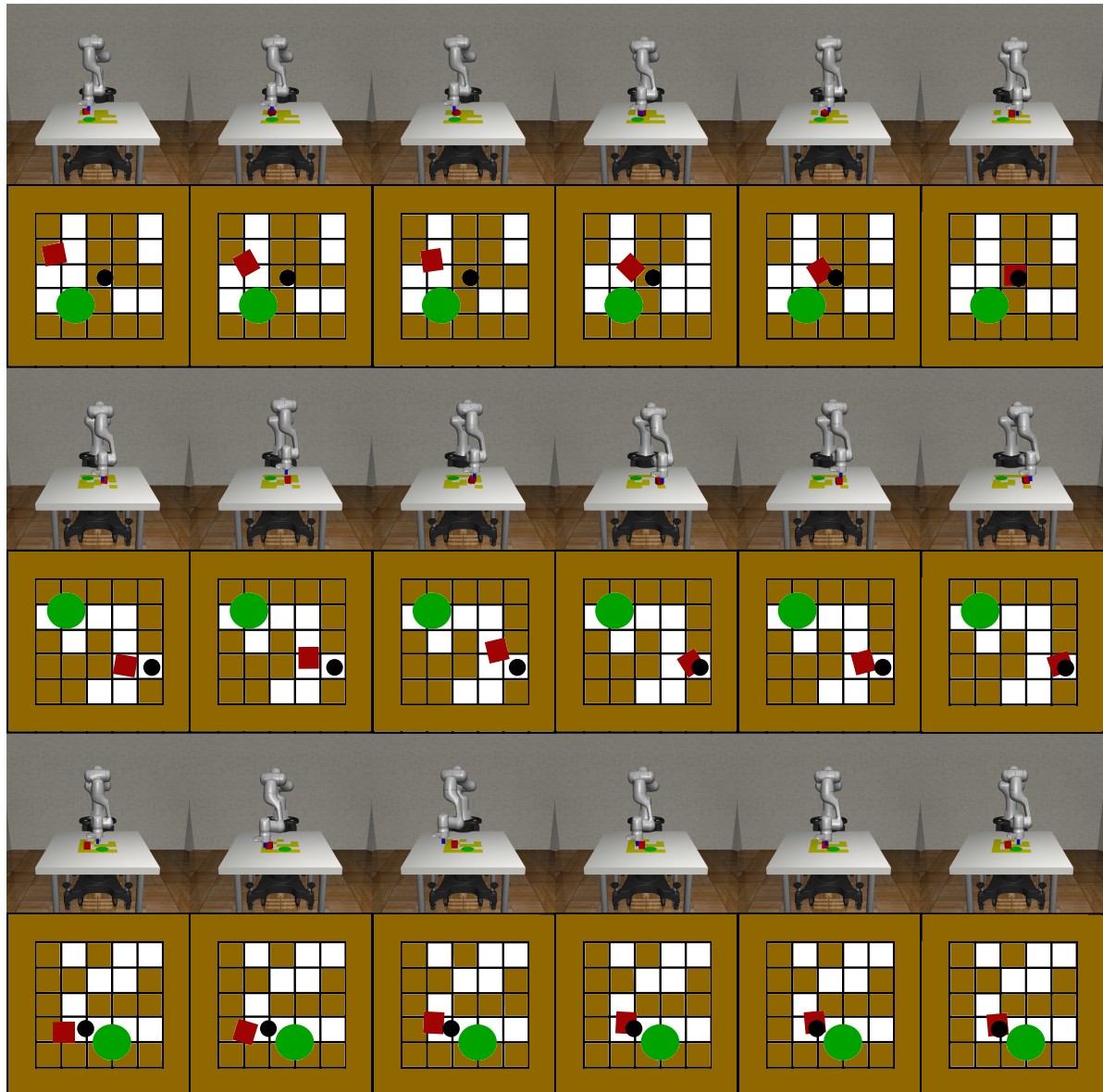

Figure 9: Visualization of the Robot Pushing Block skill, where the upper row renders the robot environment, and the lower row illustrates the state on a grid. The block goal is represented as an x,y position, since the block z coordinate is not controllable, represented on the grid as a black dot (the green circle is the task goal, which relates only to extrinsic reward.

.

We tested the model-learning component of CDL and found that it is able to detect relationships between difficult-to-model objects like the paddle and the ball or the gripper and the block. However, this takes a large number of time steps ($>500000$), and just because a relationship is detected does not mean that the model is generating useful samples for the policy. In particular, this suggests that having a general model to capture general causality is only so useful in RL tasks. Since HIntS creates a model to capture certain rare, but incredibly task-important states, and to identify those well, it ends up being more useful for performance, even though the overall model is probably worse at predicting the next state.

Finally, pairwise interaction tests are attractive in comparison to CDL because in domains with many factors (there are 100 blocks in Breakout) and thus a large state space, this can become prohibitively expensive for

CDL. Since CDL uses $P(\mathbf{x}'|\mathbf{x}/x_i)$ to detect causal relationships, where $\mathbf{x}/x_i$ denotes that the factor $x_i$ is cut from the overall state, this results in a number of models that grows in the number of objects, where each network must include all the objects. We actually change the state space of Breakout so that each block state only has one value (whether that block exists or not), because otherwise, the cost of this computation would be prohibitively expensive.

### I.3   HAC variants

HAC typically uses the full state of the environment as actions for the higher level policies and goals for the lower level policies. However, this means that the HAC higher level policies need to select complete states to reach, and the HAC low level policies have to reach those states. This is next to impossible, and that means the hierarchy ends up learning nothing: the low level states learn nothing because they cannot reach any of the goals, and the high level states learn nothing because their actions are meaningless.

We mitigate this issue by assigning HAC layers to the objects, essentially emulating what HIntS does by having each layer control an entity. In Breakout, this means that HAC has one layer that has block goals/reward and outputs ball goals, one layer with ball goals and then outputs paddle goals, one layer to convert paddle goals to action goals. However, this full hierarchy also fails because HAC relies on hindsight that is based on a fixed duration of time. Unfortunately, this often means that the ball interactions appear very rarely in the replay buffer, and the top level using actual rewards has no hindsight targets. The best we could find was with a 2-layer hierarchy that had extrinsic reward as the top level signal, and output ball velocities to a low level policy that outputs primitive actions. In this case, outputting upward velocities were somewhat learnable and would give some reward. A similar object-level hierarchy was used for robot pushing, though in this case, there was no clear reason why the layers should fail except that the reward structure is difficult enough that HAC probably needed to pre-learn the block moving policy before doing anything else. However, that would make it the same as HIntS in terms of the final structure.

### I.4   RIDE details

Rewarding impact-driven exploration (RIDE) Raileanu & Rocktäschel (2020) uses a learned forward and inverse dynamics model, combined with count-based regularization to provide an intrinsic reward for exploration. We ablated over the RIDE tradeoff parameter (the amount of weight for the intrinsic versus extrinsic reward), the RIDE learning rate (rate to train the forward/inverse models) the network architecture (changing the number of hidden layers and pointnet or MLP implementation), and the count-based scaling (rate to reduce the reward for similarly hashed components). We implemented hashing by taking the learned embedding from the forward and inverse models, applying a sigmoid and then taking the values greater than 0.5 as 1 and less as 0.

This method has been applied to visual navigation domains with some success, and in this work, we apply what appears to be the first use of this to a factorized dynamical environment. However, RIDE struggles to perform in both environments. We speculate this is because it relies on the inverse dynamics to provide a significant signal to learn a meaningful embedding. Unlike in pixel-based environments, where the embedding often retains much of the information about the state through the convolutional layers, in the factorized format much of the information can be lost, even with a pointnet, because of the use of fully connected layers. As a result, the RIDE-learned representation quickly converges to the minimal information needed to recover the action. In this case, the forward model is easy to predict, and since the loss is based on the l2 error in the prediction of the next state, without much of the information the agent quickly loses much of the intrinsic motivation. This is true in both Breakout and Robot Pushing, where there are a significant number of objects whose state does not change much (the blocks and obstacles), and whose state is not closely correlated with actions. As a result, the RIDE loss struggles to actually provide much exploration bonus. The performance of RIDE mirrors the performance of vanilla RL for the simple reason that the best choice of RIDE reward scaling (after searching through several orders of magnitude of scalings), is the one that has the least negative impact. We think that the slight negative impact might be from the intrinsic reward: in Breakout, once the good behavior has been found (bouncing the ball), the task is closer to exploitation than exploration, and overt exploration often reduces performance. For computational reasons, and because

| METHOD | NO PROXIMITY | PROXIMITY |
|--------|--------------|-----------|
| FP | $2.4 \cdot 10^{-3}$ | $2.2 \cdot 10^{-5}$ |
| FN | 0.07 | 0.003 |

Table 3: Table of breakout interaction predictive ability. A false positive (FP) is when the interaction model incorrectly predicts a ball bounce, measured per state, and a false negative (FN) is when the interaction model fails to identify a ball bounce, measured per bounce. We get these bounces from the simulator, so these comparisons are ground truth. These ball bounces are not given to the interaction model during training—we just use them for this evaluation. While both FP, FN are undesirable, FN is a greater issue for learning because they result in missed ball bounces.

| PARENT | ACT | ACT | PAD | ACT | ACT | GRP |
|--------|-----|-----|-----|-----|-----|-----|
| TARGET | PAD | BALL | BALL | GRIP | BLK | BLK |
| PE | 1.03 | 1.21 | 6.81 | 4.8 | 0.22 | 2.2 |
| AE | 3e−3 | 1.69 | 0.30 | 0.17 | 0.18 | 0.6 |

Table 4: Table of interaction test scores (Equation 9) for interaction training in breakout and Robosuite pushing between pairs of objects. Act is actions, Pad is the paddle, Grip is the gripper. PE is passive error in pixels for breakout (left) and cm in Robosuite pushing (right), AE is the active error in pixels/cm (weighted by interaction). In this case, we convert likelihoods comparisons to l2 distance in $cm$ because it is more interpretable.

there is no reason that an intrinsic reward method such as RIDE should show significantly different transfer than pretraining-fine tuning, we did not run RIDE on the Breakout variants.

| Algo | Break | Push | single | hard | big | neg | center | prox |
|---|---|---|---|---|---|---|---|---|
| Base | $85.6 \pm 8.3$ | $30 \pm 0$ | $-5.8 \pm 0.5$ | $-7.1 \pm 0.8$ | $.74 \pm 0.1$ | $2.9 \pm 0.3$ | $-37 \pm 11$ | $0.1 \pm 0.2$ |
| PT | NA | NA | $-5.4 \pm 0.9$ | $-6.1 \pm 0.4$ | $.79 \pm 0.1$ | $-7.0 \pm 19$ | $-42 \pm 9$ | $0.2 \pm 0.1$ |
| HAC | $74.3 \pm 8.2$ | $-43 \pm 89$ | NA | NA | NA | NA | NA | NA |
| RIDE | $92.5 \pm 8.2$ | $-31 \pm 2.2$ | NA | NA | NA | NA | NA | NA |
| CDL | $-49 \pm 1.6$ | $-33 \pm 1.0$ | $-10.0 \pm 0.0$ | $-9.7 \pm 1.5$ | $-9.26 \pm 2.7$ | $-49.9 \pm 2.5$ | $-49.8 \pm 0.6$ | $-10.0 \pm 0.0$ |
| C-HyPE | $76 \pm 15$ | $-21 \pm 5$ | $-6.4 \pm 3.4$ | $-7.3 \pm 1.0$ | $.35 \pm 0.05$ | $-0.72 \pm 0.9$ | $-39 \pm 10$ | $-0.6 \pm 2.5$ |
| R-HyPE | $95 \pm 3$ | $-90 \pm 19$ | $-5.6 \pm 0.45$ | $-5.1 \pm 0.49$ | $.67 \pm 0.06$ | $2.9 \pm 0.46$ | $-20 \pm 1.5$ | $-0.3 \pm 0.3$ |
| HIntS | $99 \pm 0.2$ | $-21 \pm 5$ | $-3.2 \pm 1.2$ | $-4.2 \pm 0.9$ | $.85 \pm 0.06$ | $3.6 \pm 0.3$ | $-12 \pm 4$ | $0.5 \pm 0.05$ |

Table 5: Table of the final evaluation of trained policies on Breakout variants HIntS trains a high level controller using the learned options, "FT" fine-tunes a pretrained model, and "Base" trains from scratch. The baselines are evaluated after 5m time steps, while HIntS is trained for 2m time steps. The abbreviations are the ones used in Section 5.1

| Parameter | Values | Relative performance |
|---|---|---|
| Continuous Learning Algorithm | SAC, DDPG | similar |
| Discrete Learning algorithm | DQN, Rainbow | similar |
| Learning rate | .0001-.0007 | significant if too high |
| Interaction Proximity | 5-7px, 0.5-1cm | significant if too small |
| $\epsilon_{\text{close}}$, parameter proximity | 1px-2px, 1cm-2cm | significant if too large |
| Network layers | 3-5 | Fixed across skills, Reward optimization used a larger network |
| Network width | 128-1024 | Fixed across skills, Reward optimization used a larger network |
| hindsight selection rate | 0.1-0.5 | somewhat sensitive |
| inline training rate | 100-1000 | sensitive if too low |
| relative action ratio | 0.05-0.3 | sensitive |
| Constant negative reward | $-0.1$-$-1.0$ | sensitive |
| Minimum Set size | 10 | Not particularly sensitive |
| Minimum test score (in log-likelihood) | 3 | Not particularly sensitive |
| Interaction Score Cutoff ($\epsilon_{\text{edge}}$) | 10 | Affects sample efficiency |
| Success Convergence cutoff | 0.01 | Affects sample efficiency vs final performance |

Table 6: Table of hyperparameters

