# OpenReview forum: "Granger-Causal Hierarchical Skill Discovery"
_TMLR — Rejected by TMLR_

### Review · Reviewer_oZTB · 2023-06-22

**Summary Of Contributions:**

This paper introduces a Hierarchical Reinforcement Learning algorithm for factored-state MDPs. The hierarchy of skill is learned in a way that each skill allows to control one particular factor of the factored MDP. How to identify which skills should learn what, based on what observations, is detailed in this paper as the HIntS algorithm. The core contribution of this paper is a method based on Granger-Causality that allows to detect factors that influence other factors (observing the factor helps predicting future values of another factor).

The proposed method to detect causality between the factors is then integrated in a fully-working HRL algorithm that is shown to learn in a sample-efficient way in Breakout and a robot pushing task. The learned skills are also shown to be useful for transfer across variations of the tasks.

**Audience:**

Yes

**Broader Impact Concerns:**

There does not seem to be broader impact concerns for this work.

**Claims And Evidence:**

Yes

**Requested Changes:**

All the requested changes are suggestions to strengthen the work, that I think is already acceptable as is.

- In general, some broad overview of this work and its intuition is missing. The authors seem very familiar with what they have designed, but the average reader has many things to discover to understand this work.
- For instance, the introduction should state what a "factor" is, as this word can have many meanings depending on the specific field. The concept of sequences of skills should also be better explained. Figure 1 is clear once the paper has been read, but is not understandable when the reader first looks at it.
- It is unclear how the first skill, the primitive actions, work. This seems related to the addition of a factor for primitive actions, but there is no clear explanation. Overall, I think that each level of the hierarchy should have an explicit list of what its observation space is, its action space (that is actually already detailed in the paper) and its reward function. Then, an example of a rollout (with t being the primitive actions time-steps, and maybe a view of a "stack" of skills being executed) would help too. For inspiration, see https://i.ytimg.com/vi/6uKZXIwd6M0/maxresdefault.jpg .
- Section 3.1: "We augment the state factors with non-state factors, such as xprim a factor describing primitive actions to discover interactions between primitive actions and state factors in the domain" is unclear
- We need a motivation of why assuming a factored state is not a big problem. Two papers are cited in Section 4 but not discussed. Factored states is a major hypothesis of this work, and a good argument of why such a factorization is easy to achieve would really broaden the impact of this paper.
- Section 4.3: is this a correct intuition?: we train to predict x'b given either only xb, or with both xb and xa. If having xa helps, then there is a causality link between xa and xb. If yes, then an intuition like this could be used in the abstract or introduction of the paper to quickly tell the reader what the contribution is about.

**Strengths And Weaknesses:**

Strengths:

- The paper is very well-written and proposes a sound method for learning skills.
- The proposed algorithm is elegant, seems easy to implement (it introduces two simple function approximators, easy to train from data and easy to use for prediction)
- The empirical results are strong, especially for an algorithm that makes few assumptions.

Overall, this paper has a high significance and a high potential impact.

Weaknesses:

- Some aspects of the paper could be made clearer, see requested changes.

---

### Review · Reviewer_KHmQ · 2023-06-22

**Summary Of Contributions:**

This paper aims to learn the interaction between objects in factored environments to help reinforcement learning algorithms to achieve better sample efficiency and generalization capability. Inspired by Granger causality, the authors design an interaction detector that can discover whether two objects have interaction from observational data. The evaluation of two robotic domains demonstrates that the proposed method not only has good generalization but also attains sample efficiency and final performance compared to several RL baselines. The main contributions of this paper can be summarized as (1)applying Granger causality to forward dynamics models to detect interactions. (2) designing a hierarchical RL framework by using discovered option chains.


**Audience:**

Yes

**Broader Impact Concerns:**

As far as I know, this is no ethical concerns in this work.

**Claims And Evidence:**

No

**Requested Changes:**

* 1.The most critical request is evaluating the algorithm in more complicated tasks. Though the Breakout task is intuitive enough to illustrate the idea, it is too simple to investigate the effectiveness of the proposed algorithm. I notice that the authors discussed these points in limitations. I agree that the method proposed in this paper is a good attempt at skill discovery, but I think the current quality of the evaluation is not solid enough to be published. I suggest improving the quality of the evaluation from two perspectives:
(1) use an environment that contains longer trajectories that contains more skills, for example, robot manipulation pick-and-place.
(2) use an environment that contains interaction between multiple objects


* 2.The right part of Figure 1 does not contain enough information. I find that this figure is mentioned in the very later portion of the paper (Section 4 and Section 5). Maybe the authors can make it similar to the left part of the figure to indicate the skill and actions to provide more information. Actually, I am not sure how does skill mean in the push environment, a concrete example will be helpful.

* 3.Reference can be added to each method in Section 5.1.


**Strengths And Weaknesses:**

**Strength**:

* 1.The idea of self-supervised skill discovery in hierarchical RL is an interesting topic and has great potential to deal with complex tasks that require too many low-level and long-horizon controls. Using the chain of skills is a promising direction to abstract the long trajectory and make it more efficient to learn high-level motions.

 * 2.Using interaction detection to discover skills is interesting since the interaction naturally separates long trajectories into segments with semantic meanings.

**Weaknesses**:

* 1.The algorithm seems to only consider a linear chain of skills, without considering the complex tasks that require multiple pre-conditions. It seems that the proposed algorithm can be extended to handle this by designing a more flexible hierarchical structure, but the current version does not consider it.

* 2.The evaluation of the proposed method is done on two simple tasks, i.e., Breakout and Push, where the Breakout game has three objects (paddle, ball, block) and the Push environment have two objects (gripper and block). I think the length of the chain is too short and the interaction between two objects is too easy to analyze. I believe the proposed method has the potential to deal with longer chains and interactions but more evidence of the experiment should be provided. For example, CALVIN, a benchmark that requires a long chain of skills, could be a good option.

* 3.The advantage of the proposed method in Figure 3 is not obvious.

---

### Review · Reviewer_H3W4 · 2023-06-26

**Summary Of Contributions:**

This paper introduces a hierarchical skill discovery algorithm based on a neural extension of Granger-Causality. The algorithm successively builds up a hierarchical sequence of skills, whereby each skill corresponds to controlling the agent in a lower-level action space to achieve goals in a discovered goal space. At each successive level, goals are discovered using a criterion based on the Granger-Causality score applied to rollouts based on taking random actions in the lower-level action space. A goal-conditioned policy is then trained for this new goal space, which then becomes the lower-level action space in the next iteration of the algorithm. Initially, the action space is simply the base action space of the environment. This paper further compares the performance of this algorithm, called HIntS, on variants of Atari Breakout and a robot block pushing task. They demonstrate that HIntS leads to improved sample-efficiency in these tasks, achieving a higher return after the same number of training steps.

**Audience:**

Yes

**Broader Impact Concerns:**

There are no broader impact concerns unique to this work.

**Claims And Evidence:**

No

**Requested Changes:**

Based on the weaknesses of this work discussed above, I request the following changes, before this work can be considered ready for publication:

- Incorporation of an analysis of the emergent complexity of HIntS in terms of the goals/action spaces that arise during training. Even something more qualitative would add much to the current results, which do not probe at precisely the most interesting, purported capability of this algorithm.
- The baselines section could be strengthened by clearly emphasizing the expected weaknesses of each of methods evaluated, highlighting the relative advantage of HIntS w.r.t. each of these methods.
- The authors improve the writing to be clearer in terms of making sure to provide motivation and definitions of the various mathematical notations and concepts.
- The authors should provide clearer descriptions of how they make sure they fairly compare each of the various methods.
- The authors should make Fig. 2 much clearer and reflective of their method.


**Strengths And Weaknesses:**

**Strengths**

- The method proposed appears novel.
- The authors compare to a wide array of baselines on both a discrete and continuous task space.
- The empirical results seem to show that HIntS results in improved sample efficiency.

**Weaknesses**

- The biggest shortcoming is that the authors do not visualize any of the emergent complexity discovered by their method, which they argue throughout the paper is the reason for its effectiveness. The authors should provide analysis of the layers of skills discovered by their method throughout training in each of the two domains studied. This would provide currently missing support for the motivation of the algorithm presented in the paper.
- The motivation of HIntS is similar to that of intrinsic motivation methods based on empowerment [1], which seek transitions that are maximally controllable by the agent’s action, e.g. based on a mutual information between action and future states. However, this work does not compare to any intrinsic motivation methods. In many ways, the goal-based formulation of HIntS effectively provides a similar “intrinsic motivation” signal, so I believe such a comparison would be warranted---perhaps to a method like RiDE [2].
- The clarity of writing throughout the manuscript can be improved significantly, e.g. many terms are used in Equations before they are defined, and there are at times somewhat uncommon notations that are used without explanation, like the notation $f(x_b)[x’_b]$. My interpretation of this notation is that this notation indicates a probability (Equation 3), while $f(x_b)$ refers to the actual predicted value of $x’_b$. However, nowhere is this important usage explained. Similarly, the $L$ terms are not defined in Equation 7 until Section 4.3, and only in passing.
- The authors also make use of the term “actual causality,” citing Halpern, 2016, but this comment can be quite cryptic to readers. It should be expounded upon if it is indeed important for understanding the benefits of the method proposed.
- Likewise, Figure 2 is quite confusing, and I found it hard to make sense of what it was trying to communicate, even after reading the paper in detail. Ideally, Figure 2 should provide a system overview of the HIntS algorithm that clearly shows the key steps of the algorithm, and importantly, how the successively discovered goals become the hierarchical action space in the next iteration of the algorithm.
- The experimental setup is not clearly explained. Importantly, HIntS makes use of distinct  rollout phases throughout each iteration, consisting of collecting interactions and option learning. Similarly, CMA-based methods like CMA-ES-HyPE makes use of a population. The authors should explain how these differences in rollout collection are made consistent in their experimental comparisons, which list 2M as the time step budget in each setting, but what this time budget corresponds to for each method is unclear. It is important to convince the reader that the method for setting the time step budgets across methods makes for a fair comparison.

**References**

[1] Mohamed, Shakir, and Danilo Jimenez Rezende. "Variational information maximisation for intrinsically motivated reinforcement learning." Advances in neural information processing systems 28 (2015).

[2] Raileanu, Roberta, and Tim Rocktäschel. "RIDE: Rewarding Impact-Driven Exploration for Procedurally-Generated Environments." International Conference on Learning Representations.

---

### Decision · Action_Editors · 2023-08-01

**Recommendation:** Reject

**Comment:**

This paper proposes a hierarchical RL algorithm that discovers chain structures in decision making using Granger causality. The main concerns raised by the reviewers were:

* **Over-claiming:** If the paper claims to do a hierarchy discovery but ultimately does a chain, this needs to be clarified. Please add a paragraph to the paper to make it clear. The earlier the better.

* **Insufficient experiments:** I consider the experiments sufficient.

* **Paper is unclear at times:** I checked the paper and this is my main concern. In particular, the technical writing is subpar for a machine learning paper and needs to be updated in a major way before the paper is reviewed again.

In addition to the comments of the reviewers, these are my comments on Section 3.1. This section is barely a quarter of a page long. My comments illustrate how the writing can be improved:

  * $\mathcal{X}$ is undefined. Say that this is the state space.

  * $\mathcal{X}_o$ is undefined. Say that this is the initial set of states.

  * How can $x_o \in \mathcal{X}_o$ be a distribution?

  * Transition function should be defined as $T: \mathcal{X} \times A_{prim} \times \mathcal{X} \to [0, 1]$.

  * Reward function should be defined as $R: \mathcal{X} \times A_{prim} \times \mathcal{X} \to \mathbb{R}$.

  * The factored state should be $x = (x_1, \dots, x_n)$.

  * The factored state space should be $\mathcal{X} = \mathcal{X}_1 \times \dots \times \mathcal{X}_n$.

  * The only citation in the section should be done using \\citep.

  * $x^{(t)}$ and $a_{prim}^{(t)}$ are undefined.

  * $\mathcal{R}$ and $R_i$ are undefined.

  * It is not clear what $\mathcal{X}_c irc^i \in \mathcal{X}_o$ means.

I strongly encourage the authors to resubmit after the writing, including mathematics, is improved.

**Audience:**

Yes. This paper would be of a general interest to RL and robotics communities.

**Claims And Evidence:**

Yes. The proposed method is evaluated on two RL domains: Breakout and Robot Pushing. Reviewer KHmQ would like to see more domains but I believe that the current contribution is sufficient. The authors need to clarify that they focus on discovering a chain of skills as opposing to general hierarchies.

**Resubmission Of Major Revision:**

The authors may consider submitting a major revision at a later time.